# Routes of Zika virus dissemination in the testis and epididymis of immunodeficient mice

Konstantin A. Tsetsarkin[1], Olga A. Maximova[1], Guangping Liu[1], Heather Kenney[1], Natalia Teterina[1], Marshall E. Bloom[2], Jeffrey M. Grabowski[2], Luwanika Mlera[2], Bianca M. Nagata[3], Ian Moore[3], Craig Martens[4], Emerito Amaro-Carambot[5], Elaine W. Lamirande[5], Stephen S. Whitehead [5] & Alexander G. Pletnev[1]

Sexual transmission and persistence of Zika virus (ZIKV) in the male reproductive tract (MRT) poses new challenges for controlling virus outbreaks and developing live-attenuated vaccines. To elucidate routes of ZIKV dissemination in the MRT, we here generate microRNA-targeted ZIKV clones that lose the infectivity for (1) the cells inside seminiferous tubules of the testis, or (2) epithelial cells of the epididymis. We trace ZIKV dissemination in the MRT using an established mouse model of ZIKV pathogenesis. Our results support a model in which ZIKV infects the testis via a hematogenous route, while infection of the epididymis can occur via two routes: (1) hematogenous/lymphogenous and (2) excurrent testicular. Co-targeting of the ZIKV genome with brain-, testis-, and epididymis-specific microRNAs restricts virus infection of these organs, but does not affect virus-induced protective immunity in mice and monkeys. These defined alterations of ZIKV tropism represent a rational design of a safe live-attenuated ZIKV vaccine.

---

[1] Laboratory of Infectious Diseases, National Institute of Allergy and Infectious Diseases (NIAID), National Institutes of Health (NIH), Bethesda 20892-3203 MD, USA. [2] Biology of Vector-Borne Viruses Section, Laboratory of Virology, Rocky Mountain Laboratories, NIAID, NIH, Hamilton 59840 MT, USA. [3] Infectious Disease and Pathogenesis Section, Comparative Medicine Branch, NIAID, NIH, Rockville 20892 MD, USA. [4] Research Technologies (RT) Section, RT Branch, Rocky Mountain Laboratories, NIAID, NIH, Hamilton 58940 MT, USA. [5] Laboratory of Viral Diseases, NIAID, NIH, Bethesda 20892-3210 MD, USA. Correspondence and requests for materials should be addressed to A.G.P. (email: apletnev@niaid.nih.gov)

Zika virus (ZIKV) is a mosquito-borne flavivirus that was initially reported in Africa, but now is causing outbreaks in Asia and Latin America[1,2]. The majority of ZIKV infections are asymptomatic or associated with mild illness. However, significant developmental abnormalities were reported in babies born from mothers who contracted ZIKV during pregnancy (reviewed in ref. [3,4]). In addition, ZIKV is capable of establishing long-term persistent infection in humans and utilizing non-vector transmission routes. This includes sexual transmission from symptomatic or asymptomatic sexual partners, and after a partner's recovery from ZIKV illness[1,2,5]. Infectious virus has been isolated from human semen, and ZIKV RNA can persist in semen for up to 1 year after onset of ZIKV illness[6–12] (reviewed in ref. [13,14]). The long-term viral persistence in male reproductive tract (MRT) and the newly described route of transmission pose new challenges for controlling ZIKV outbreaks and for the development of a safe live-attenuated ZIKV vaccine.

Mechanisms of ZIKV dissemination and persistence in the MRT are poorly understood. In monkeys, ZIKV was detected in the testes, prostate, and seminal vesicles and can persist for at least 28 days post infection (dpi)[15]. Human testicular tissue explants support active ZIKV replication, which occurs primarily in macrophages and germ cells[16]. In immunodeficient mice, ZIKV primarily targets the testis and epididymis, and to a lesser extent, the prostate and seminal vesicles[17–21]. In this study, we use a host microRNA (miRNA)-targeting approach to trace routes of ZIKV dissemination in the MRT of mice. As ZIKV generated in one part of the MRT can be transported to the other parts located downstream of excurrent flow, we focus only on the ZIKV interaction with major upstream organs, namely, the testis and epididymis.

## Results

**Model to study tissue tropism of miRNA-targeted ZIKVs.** The ZIKV-NS3m infectious cDNA clone of ZIKV [strain Paraiba_01/2015; isolated in Brazil[22]] was modified by inserting two copies of identical 20 nucleotides (nt) scramble (scr) sequences after nt 8 and 14 of the 3′NCR of ZIKV genome generating 2×scr virus (Fig. 1a, Supplementary Fig. 1). ZIKV-NS3m and 2×scr were used to intraperitoneally (ip) infect adult (4–6 week-old) AG129 male mice at a dose of $10^6$ pfu. The AG129 mice are deficient for type I and type II interferon receptor genes, which makes them highly susceptible to ZIKV infection[23–25]. All mice infected with 2×scr succumbed to disease within 15 dpi, which occurred ~ 3.5 days later compared with ZIKV-NS3m (Fig. 1b). Both viruses replicated efficiently in the serum and spleen of mice at 1 and 3 dpi, followed by a substantial reduction in virus titers (Fig. 1c, d). In the brain, 2×scr replicated slower compared with ZIKV-NS3m (Fig. 1e) and reached maximum titers ~ 12 and 15 dpi (Fig. 1e). In the testis and epididymis, replication of 2×scr was also somewhat attenuated compared with ZIKV-NS3m during the late course of infection (Fig. 1f, g). Notably, in contrast to virus replication kinetics in the serum, spleen, and brain, two peaks of 2×scr virus growth were apparent in the testis at 3 and 12 dpi (Fig. 1f) and epididymis at 3 and 9 dpi (Fig. 1g). Based on these observations, we focused on the two time points (3 and 12 dpi) for future evaluation of infection with all miRNA-targeted ZIKV clones. Sequencing analysis of the 2×scr virus that was isolated from the spleen (3 dpi; $n = 3$), brain (12 dpi; $n = 7$), testis (12 dpi; $n = 7$), and epididymis (12 dpi; $n = 4$) showed that the inserted sequences remained intact.

**Inhibition of ZIKV replication in the seminiferous tubules by miRNA targeting.** Tissue-specific expression profiles of host miRNAs provides a convenient strategy to study virus tropism

and to attenuate viral pathogenesis in selected organs[26,27]. To test whether miRNA-targeting can restrict ZIKV replication in mouse testis, we replaced scr sequences in 2×scr with sequences complementary to testis-specific miRNAs mir-202–5p and mir-449a-5p [Fig. 2a[28–30]], generating 2×202(T) and 2×449a(T) viruses, respectively (Fig. 2b, Supplementary Fig. 1). To validate organ specificity of miRNA targeting, we generated two additional constructs with targets for the central nervous system (CNS)-specific mir-124–3p and mir-9–5p (Fig. 2a, Supplementary Fig. 1)[31]. All miRNA-targeted viruses grew with similar kinetics in Vero cells (Fig. 2c), indicating that inserted sequences did not affect ZIKV replication fitness in the cells that do not express the above-mentioned miRNAs.

miRNA targeting of ZIKV genome attenuates virus replication in the cells expressing given miRNA(s). This attenuation can manifest as a reduction of the miRNA-targeted virus titer in respective tissues and/or by an emergence of mutations within miRNA targets in viral genomes (escape mutants). To account for both phenomena, AG129 mice were infected ip with $10^6$ pfu of miRNA-targeted viruses (2×scr, 2×9(T), 2×124(T), 2×202(T), and 2×449a(T); Fig. 2b), and samples derived from mouse organs were used to determine viral titer by plaque assay. This was followed by sequence analysis of virus isolates from the tissues of interest to evaluate the stability of miRNA targets.

Replication of each miRNA-targeted virus in the serum of AG129 mice at 1 dpi and in the spleen at 3 dpi was indistinguishable from that of the 2×scr virus (Fig. 2d, e). At 3 dpi, all viruses remained stable in the spleens dissected from 100% of mice ($n = 3$, for each virus), indicating that inserted sequences do not affect ZIKV ability to induce a systemic infection in mice. The titers of all miRNA-targeted viruses in the testis at 3 dpi were also comparable to that of 2×scr (Fig. 2f), and all viruses remained genetically stable. Moreover, with exception of 2×202(T), all miRNA-targeted viruses isolated from mouse testes at 12 dpi also remained stable, reaching a titer similar to that of 2×scr virus (Fig. 2g). The testicular samples from 2×202(T)-infected mice were divided into two groups (Te-2×202/(T)-stb and Te-2×202(T)-mut) based on stability of the inserted miRNA target sequences (stb—stable, or mut—mutated miRNA target). Only 50% of mice ($n = 12$) infected with 2×202(T) retained intact inserted sequences in their testes at 12 dpi, whereas in the remaining mice, the testis-derived viruses acquired deletions that affected both mir-202–5p targets (Supplementary Table 1). Viral load in the testes of Te-2×202(T)-stb mice was ~10,000 fold lower compared to that of 2×scr. Importantly, viral load in the testes of Te-2×202(T)-mut mice was only ~10-fold lower than that of 2×scr-infected mice, but was ~ 1000 fold higher compared with that of Te-2×202(T)-stb mice (Fig. 2g). Insertion of targets for mir-449a-5p into the ZIKV genome did not affect virus replication in the testis, and the virus was excluded from further analysis.

To validate organ specificity of miRNA-mediated attenuation of ZIKV, we compared titers of miRNA-targeted viruses in the brain of mice at 12 dpi. Neither mouse nor human CNS express mir-202–5p[28,29]. Expectedly, replication of 2×202(T) in the brains of Te-2×202(T)-stb and Te-2×202(T)-mut mice was indistinguishable from that of 2×scr virus. In contrast, titers of viruses that carried targets for the CNS-specific miRNAs (2×124(T) and 2×9(T)) were significantly reduced (Fig. 2h). Importantly, escape mutations in the mir-202–5p targets were not detected in the 2×202(T) virus isolated from the brains of all Te-2×202(T)-stb and Te-2×202(T)-mut mice (Supplementary Table 1).

To determine which step of testicular infection is affected by mir-202–5p targeting, we compared the growth kinetics of 2×202(T) and 2×scr viruses in the testis. Replication of both viruses was undistinguishable during the early phase of testicular infection

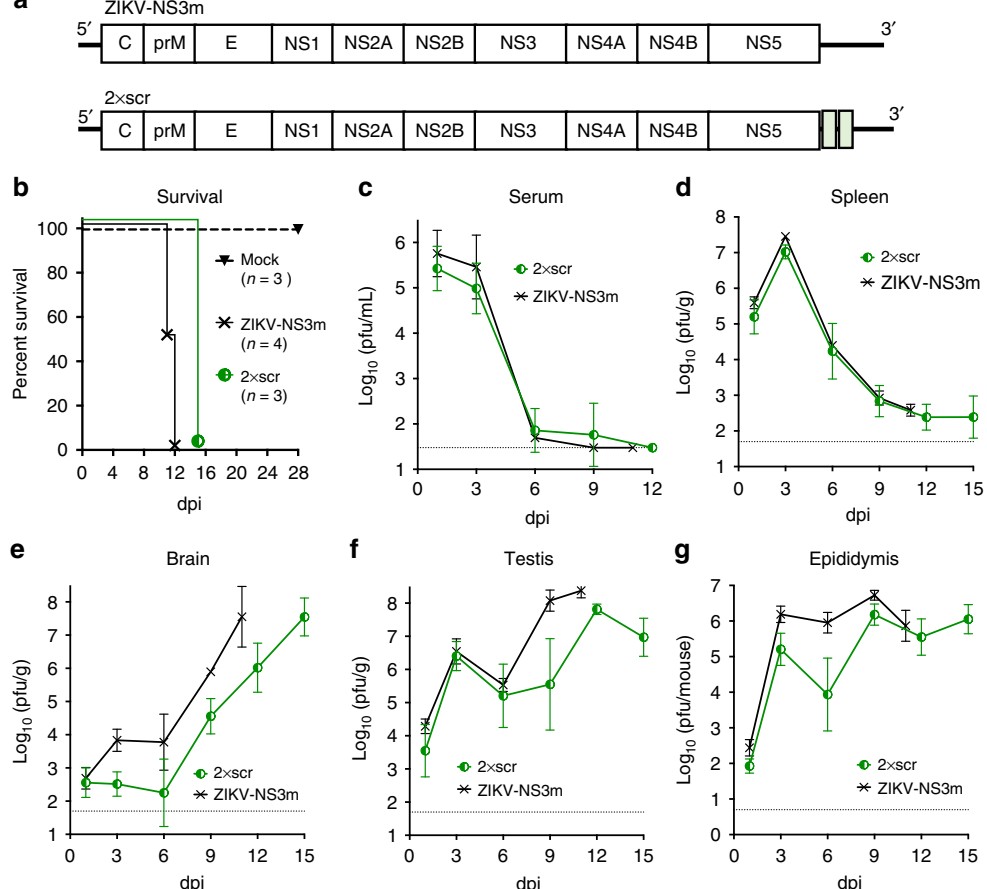

**Fig. 1** Model to study the effects of miRNA targeting on ZIKV tissue tropism and pathogenesis. **a** Schematics of ZIKV-NS3m and 2×scr viral genomes. Green boxes indicate position of scr sequence insertions. **b** Survival of AG129 mice inoculated ip with $10^6$ pfu of ZIKV-NS3m, 2×scr virus or with a diluent (mock). **c–g** Growth kinetics of ZIKV-NS3m and 2×scr viruses in the serum **c**, spleen **d**, brain **e**, testis **f**, or epididymis **g** of adult AG129 male mice inoculated ip with $10^6$ pfu of virus. Mean viral load in the serum or organ homogenates ($n = 3$–7 per time point) ± standard deviation (SD; shown as error bars) was determined by titration in Vero cells. The dashed lines indicate the limit of virus detection: 1.5 $\log_{10}$(pfu/mL) for serum **c**, 1.7 $\log_{10}$(pfu/g) for spleen, brain, or testis **d–f**, and 0.7 $\log_{10}$(pfu/mouse) for epididymis **g**

(0–6 dpi), but the titer of 2×202(T) substantially decreased at later time points (9 and 12 dpi) (Supplementary Fig. 2). Immunohistochemical examination of ZIKV antigen distribution in the testes revealed that at 3 dpi, both 2×scr and 2×202(T) primarily infected cells residing in the testicular interstitium (Fig. 3b, c). By 12 dpi, both viruses were cleared from the interstitium (Supplementary Fig. 3), but only 2×scr invaded seminiferous tubules (Fig. 3e). In contrast, ZIKV antigen was not detected in seminiferous tubules in 83% of analyzed testes ($n = 6$) from mice infected with 2×202 (T) virus (Fig. 3g), indicating that mir-202–5p targeting only affected the ability of ZIKV to replicate within the cells of seminiferous tubules during the later stage of infection. This is consistent with our in situ hybridization data on expression of mir-202–5p in the cells located within the seminiferous tubules (Supplementary Fig. 4). ZIKV antigen was observed in one of the six testes infected with 2×202(T) virus (Supplementary Fig. 5), that likely reflects accumulation of escape mutation(s) in the mir-202–5p target sites (Supplementary Table 1).

**ZIKV can invade epididymis, bypassing the testis**. A specific inability of 2×202(T) to replicate in cells of seminiferous tubules and low expression of this miRNA in the epididymis (Fig. 4a, Supplementary Fig. 4) provides a model for investigation of the potential routes of ZIKV spread within the MRT. Spermatozoa produced in seminiferous tubules of the testis are transported via

efferent ducts to the epididymis where they are stored and undergo maturation. To determine whether ZIKV utilizes this route to invade the epididymis, we compared titers of 2×scr and 2×202(T) in the epididymis during early (3 dpi) and late (12 dpi) infection (Fig. 4). At 3 dpi, both viruses replicated to similar titers, remained stable, and infected cells residing in the epididymal interstitium (Fig. 4e, f). However, in contrast to the miRNA-mediated suppression of 2×202(T) virus replication in the testis at 12 dpi, this virus was able to freely replicate in the epididymis of Te-2×202(T)-stb mice, similar to the 2×scr (Fig. 4c). Sequence analysis confirmed the mir-202–5p target stability in the epididymis samples isolated from all Te-2×202(T)-stb mice ($n = 5$, Supplementary Table 1). Immunohistochemical examination showed that at 12 dpi, both 2×202(T) and 2×scr viruses were cleared from the interstitium (Supplementary Fig. 6) and both viruses progressed into the epididymal epithelium (Fig. 4h, j). This finding, together with the lack of correlation between testicular and epididymal titers of 2×202(T) virus in the Te-2×202(T)-stb mice (Fig. 2g, 4c), suggests that ZIKV testicular infection is not a prerequisite for downstream epydimal infection.

Sequence analysis of the 2×202(T) virus derived from the epididymis of Te-2×202(T)-mut mice at 12 dpi showed no mutations in miRNA target sequences in four of the five samples, additionally supporting the hematogenous (testis-independent ZIKV) invasion of the epididymis (Supplementary Table 1).

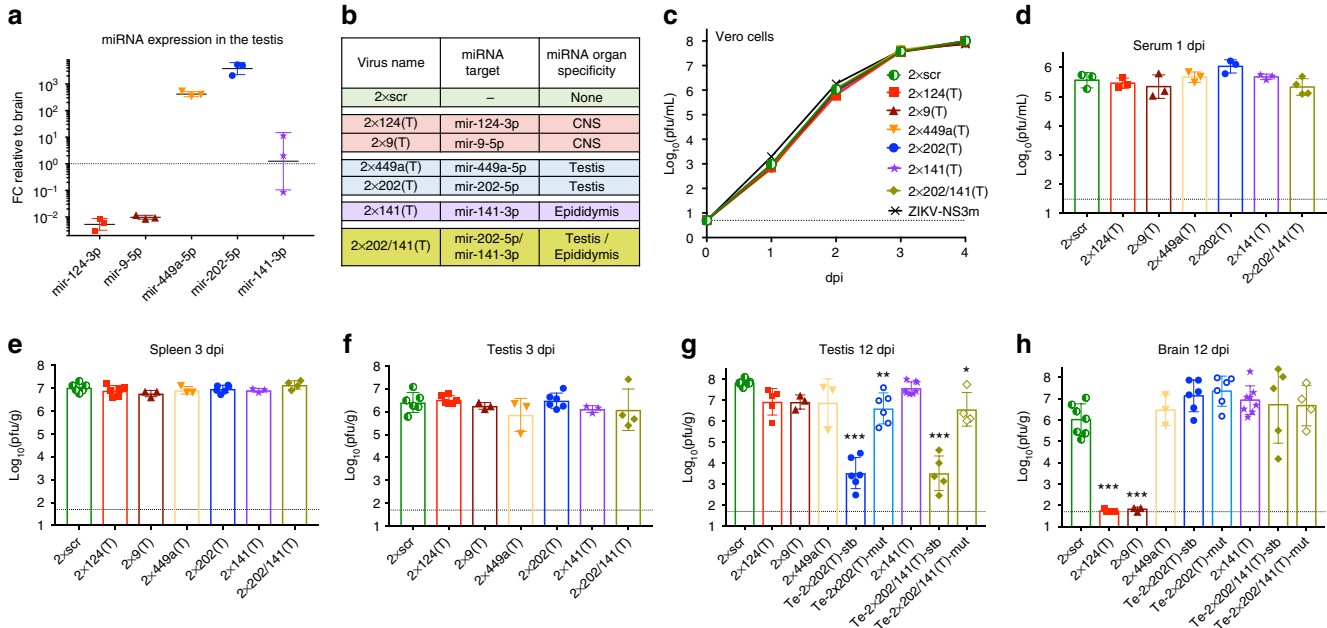

**Fig. 2** Effect of miRNA target insertions on ZIKV replication in the serum and organs of AG129 mice. **a** Relative expression of selected miRNAs in the testis compared with their expression in the brain. Data indicate the ratio (folds change [FC]) of normalized counts for miRNAs isolated from the testis to those isolated from the brain of individual AG129 mice ($n = 3$). Horizontal line and error bar are geometric mean and geometric SD, respectively. **b** List of viruses with inserted miRNA targets. **c** Growth kinetics of recombinant viruses in Vero cells. Results are presented as an average titer of two biological replicates ± SD (shown as error bars). **d–h** Adult AG129 mice were infected ip with $10^6$ pfu of 2×scr or miRNA-targeted viruses and were killed at 3 or 12 dpi. Mean viral titer ± SD in the serum at 1 dpi **d**, spleen at 3 dpi **e**, testis at 3 dpi **f**, and 12 dpi **g**, and brain at 12 dpi **h** was determined by titration in Vero cells. Mice infected with 2×202(T) and 2×202/141(T) viruses were divided into two groups (stable, stb and mutant, mut) based on the miRNA target sequence stability in the testis at 12 dpi. The titers of 2×202(T) or 2×202/141(T) viruses in the brain and testis in these two groups of mice are presented separately. The dashed lines indicate the limit of virus detection: 0.7 $\log_{10}$(pfu/mL) for Vero cells (**c**), 1.5 $\log_{10}$(pfu/mL) for serum **d** and 1.7 $\log_{10}$(pfu/g) for spleen, brain, or testis **d–h**. Differences between the titer of 2×scr and the titer of each of miRNA-targeted virus in mouse serum or organs were compared using one-way ANOVA (***$p < 0.001$, **$p < 0.01$, *$p < 0.05$). Differences between the replication of 2×scr and other viruses in Vero cells were compared using two-way ANOVA, and were not statistically significant ($p > 0.05$)

Sequence of remaining 2×202(T) isolate from the epididymis of Te-2×202(T)-mut mouse (see mouse# 12 in the Supplementary Table 1) identified a minor population of ZIKV genomes, which presented as minor signals in sequencing electropherogram (Supplementary Fig. 7). The location of this minor heterogeneity coincides with the site of escape deletion that was observed in the testicular ZIKV isolate in the same animal, suggesting that ZIKV can also be transported to the epididymis via excurrent ducts from infected testis.

**Excurrent testicular route of ZIKV infection of epididymis.** The ability of ZIKV to utilize a hematogenous/lymphogenous route to infect the epididymis complicates examination of the excurrent testicular route of dissemination. To overcome this constraint, we aimed to develop a miRNA-targeted virus, which is specifically restricted in its ability to replicate in the epididymal epithelium. We selected the mir-141–3p, which is highly expressed in the epididymis (Fig. 4a), but not in the testis or brain (Fig. 2a)[28,32–36]. Mir-141–3p is a member of mir-200 family, which regulates maintenance of epithelial cell phenotype[37–39] and is predominantly expressed in the epididymal epithelium (Supplementary Fig. 4f).

We generated 2×141(T) virus that contained 2 targets for mir-141–3p miRNA in the 3′NCR (Supplementary Fig. 1). Replication of 2×141(T) was similar to 2×scr in Vero cells, mouse serum, spleen, testis, and brain (Fig. 2c–h). Sequence analysis confirmed that 2×141(T) virus isolated from the spleen at 3 dpi, and from the brain or testis at 12 dpi, remained genetically stable (Supplementary Table 2). Histological examination of testes from

2×141(T)-infected mice at 12 dpi did not reveal any differences in ZIKV antigen distribution compared to 2×scr-infected testes (Fig. 3h), indicating that targeting for mir-141–3p does not selectively alter ZIKV ability to establish systemic infection or to replicate in the brain and testis.

Surprisingly, the titer of 2×141(T) virus in the epididymis at 3 and 12 dpi was not significantly different when compared to that of 2×scr (Fig. 4b, c). However, sequence analysis revealed that at 12 dpi, 20% (one out of five) of epididymal samples contained a mixed population of 2×141(T) genomes with intact miRNA target region and with a 3′NCR deletion that affects both mir-141–3p targets (Supplementary Fig. 8, Supplementary Table 2). This suggests the existence of miRNA-mediated selective pressure against replication of the 2×141(T) virus in the epididymis. Immunohistochemical analysis at 12 dpi showed that the 2×141 (T) did not replicate in the epididymal epithelium in the 75% of samples ($n = 4$) (Fig. 4k), suggesting that ZIKV-targeting for mir-141–3p selectively restricts viral replication in the epididymal epithelium. Since all of the testes ($n = 4$) isolated from the same 2×141(T)-infected mice stained positive for ZIKV antigen (Fig. 3h), we speculated that 2×141(T) could be transported to the epididymis from infected testes in a cell-free form. This could explain the observed high viral titer despite the absence of ZIKV antigen in epididymal epithelium of 2×141(T)-infected mice at 12 dpi. We also noticed many rounded, ZIKV antigen positive cells/ cell debris within the epididymal lumen of 2×141(T)-infected mice (Fig. 4k). While this manuscript was under consideration, a new study identified these ZIKV positive epididymal luminal cells as spermatids (Prm2 + ), which have been sloughed from the

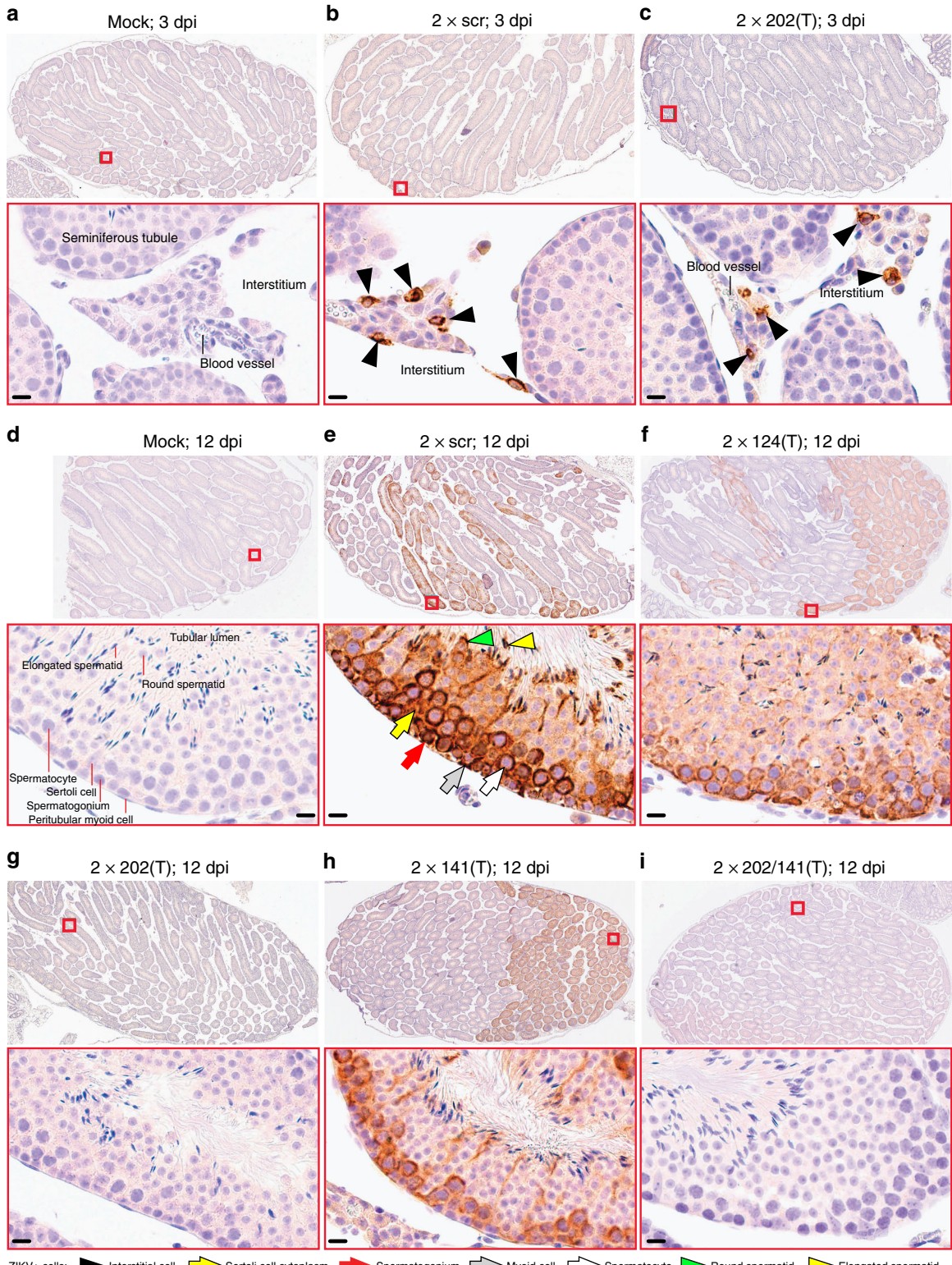

**Fig. 3** ZIKV genome targeting for mir-202–5p restricts virus ability to infect cells of the seminiferous tubules. Adult AG129 mice were infected ip with $10^6$ pfu of 2×scr, various miRNA-targeted viruses, or mock-inoculated and killed at 3 dpi **a–c** or 12 dpi **d–i**. Representative images of ZIKV antigen distribution in the testes of mice that were mock-inoculated or infected with indicated viruses are shown on the indicated dpi (2–6 mice per group). ZIKV antigen in a whole testis are shown in the top panel. Red boxes indicate the areas which are presented at higher magnification images on the bottom of each panel. Testicular structural elements are indicated in **a–c**. The cellular elements of seminiferous tubules are indicated in **d**. Labeling used within panels is indicated at the bottom of the figure. Scale bars: 10 μm

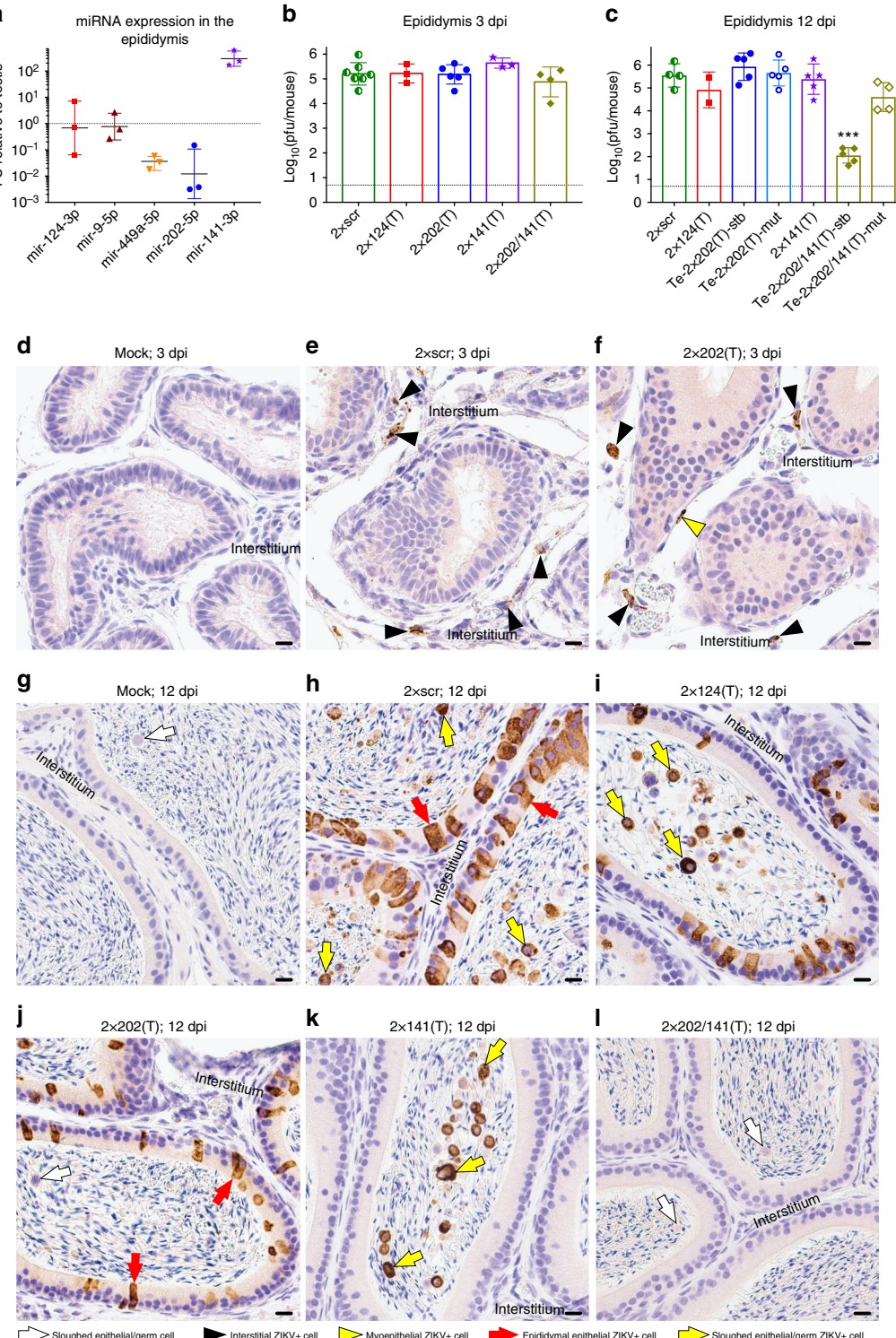

**Fig. 4** Replication of the miRNA-targeted ZIKVs in the epididymis of AG129 mice. **a** Relative expression of selected miRNAs in the mouse epididymis compared with their expression in the testis. Data show the ratio (FC) of normalized counts for miRNAs isolated from the epididymis to those isolated from the testis of AG129 mice ($n = 3$). Horizontal lines and error bars are geometric means and geometric SDs, respectively. **b–k** AG129 mice were infected ip with $10^6$ pfu of 2×scr or miRNA-targeted viruses and were killed at 3 or 12 dpi. **b** and **c** Mean viral titers ± SD (shown as error bars) in the epididymis of mice at 3 dpi **b** and 12 dpi **c**. Epididymides dissected from mice infected with 2×202(T) or 2×202/141(T) at 12 dpi were divided into two groups based on the miRNA target sequence stability (see Fig. 2 for details). The dashed line indicates the limit of virus detection (0.7 $\log_{10}$(pfu/epididymis)). Differences between the titers of 2×scr and each of miRNA-targeted viruses in the epididymis were compared using one-way ANOVA (***$p < 0.001$). **d–l** Representative images of ZIKV antigen distribution in the epididymides of mice that were mock-inoculated or infected with indicated viruses are shown on the indicated dpi (2–6 mice per group). The epididymal interstitium is indicated. Labeling used within panels is indicated at the bottom of the figure. Scale bars: 10 μm

testicular seminiferous tubules and transported to the epididymis via excurrent ducts or/and they could be infected luminal leukocytes[21].

To investigate a relative contribution of two plausible routes of ZIKV dissemination into the epididymis (hematogenous and via excurrent ducts from infected testis), we modified the 2×202(T) virus by inserting targets for mir-141–3p after the 3′-end of each of the two mir-202–5p target sequences, generating 2×202/141(T) virus (Fig. 2b, Supplementary Fig. 1). Replication of 2×202/141 (T) was indistinguishable from 2×scr in Vero cells, mouse serum at 1 dpi, spleen, testis, and epididymis at 3 dpi, and brain at 12 dpi (Fig. 2, 4). All virus isolates from brains at 12 dpi ($n = 9$) remained stable (regardless of whether the virus was stable in the testis of the same mouse; Supplementary Table 3). These results indicated that the combined insertion of mir-141–3p and mir-202–5p targets did not result in non-specific attenuation of ZIKV in mouse tissues that do not express mir-141–3p and mir-202–5p.

Next, we evaluated viral loads in the testis and epididymis of 2×202/141(T)-infected mice at 12 dpi. We divided testicular samples from 2×202/141(T)-infected mice into two groups (Te-2×202/141(T)-stb and Te-2×202/141(T)-mut) based on the stability of introduced miRNA target sequences. Importantly, the titers of the 2×202/141(T) virus in the testes and epididymis dissected from Te-2×202/141(T)-stb mice were significantly reduced compared with that of 2×scr (Fig. 2g, 4c). Immunohistochemical examination of the testis and epididymis ($n = 4$) dissected from the 2×202/141(T)-infected mice revealed a complete absence of ZIKV antigen within the seminiferous tubules (Fig. 3i), epididymal epithelium, and epididymal lumen (Fig. 4l), which is in agreement with the dual route mechanism for ZIKV invasion of epididymis.

The 2×202/141(T) virus titers in the epididymis and testis of Te-2×202/141(T)-mut mice were similar (or slightly reduced) compared with those of 2×scr (Fig. 2g, 4c). Importantly, the size and location of the deletions in the 2×202/141(T) genome were always identical in testis- and epididymis-derived viruses isolated from the same animal. However, the exact borders of deletions varied between viruses isolated from different Te-2×202/141(T)-mut mice. These deletions always eliminated or modified sequences for both mir-202–5p targets, however, they often preserved one intact mir-141–3p target located at the 3'-end (Supplementary Table 3). This strongly suggests that these deletions occurred during virus replication in the testis, but not in the epididymis, and that virus was transported to epididymis from the testis.

To rule out the possibility that testis/epididymis-specific attenuation of 2×202/141(T) was owing to a non-specific effect of long-length insertions at the 3'-NCR, we replaced all miRNA targets in 2×202/141(T) with a 99 nt sequence (Supplementary Fig. 9a) from the NanoLuc® Luciferase gene. The resulting scr (Long) virus remained stable in the testis and epididymis of 100% of mice at 12 dpi and its titer was significantly higher than the titer of 2×202/141(T) in these organs isolated from Te-2×202/141 (T)-stb mice (Supplementary Fig. 9b, 9c). However, scr(Long) replicated to nearly the same level as 2×202/141(T) in the brain of these animals (Supplementary Fig. 9d).

It is still possible that attenuation of the 2×202/141(T) virus in the epididymis was not miRNA-mediated, but was attributed to an unknown phenomena associated with a particular sequence inserted into the 3'NCR of the 2×202/141(T). To rule out this possibility, we engineered and characterized four additional viruses that contain targets for the testis- (mir-465a-3p) or epididymis-specific (mir-200c-3p) miRNAs alone or in combination with mir-202–5p and mir-141–3p (see Supplementary Fig. 10, Supplementary Tables 4 and 5 for the description of these results). A remarkable similarity between attenuational

profiles of these new viruses in the testis and epididymis with those of 2×202(T), 2×141(T) and 2×202/141(T) viruses, reinforces the conclusion that ZIKV invades epididymis simultaneously using both the hematogenous/lymphogenous and excurrent testicular routes of dissemination.

**Development of miRNA-targeted live-attenuated ZIKV vaccine.** To test whether viral genome targeting for both mir-141–3p and mir-202–5p might increase safety profile of a live-attenuated ZIKV vaccine candidate, we generated a C/3′NCR-mir(T), which simultaneously expresses a combination of targets for CNS- (mir-124–3p and mir-9–5p), testis- (mir-202–5p), and epididymis-(mir-141–3p) specific miRNAs in the duplicated capsid gene region (dCGR)[40] and in the 3′NCR (Fig. 5a, Supplementary Fig. 11). All miRNA targets of the C/3′NCR-mir(T) remained stable after 10 passages in Vero cells. To validate specificity of miRNA-mediated attenuation, we replaced miRNA targets in C/3′NCR-mir(T) virus with random/scramble sequences, generating C/3′NCR-scr virus (Fig. 5a, Supplementary Fig. 12). Both viruses attained similar titers in mouse serum at 1 dpi and in the spleen, testis and epididymis at 3 dpi (Fig. 5b–e). We also generated the 3′NCRΔ20 by inserting a well characterized attenuating 20 nt deletion[41,42] into 3′NCR of ZIKV-NS3m. Importantly, replication of C/3′NCR-mir(T) was significantly reduced compared to C/3′NCR-scr and 3′NCRΔ20 viruses in both the testis and epididymis of mice at 12 dpi (Fig. 5f, g). The titer of C/3′NCR-scr in the mouse CNS at 12 dpi was similar to that of C/3′NCR-mir(T) (Fig. 5h), indicating that non-miRNA-mediated attenuation of ZIKV owing to dCGR formation in a combination with insertion of heterologous sequences into the 3′NCR is sufficient to restrict ZIKV neuroinvasiveness.

We also compared the ZIKV-specific neutralizing antibody (NA) response after immunization of AG129 mice with C/3′NCR-mir(T) or C/3′NCR-scr viruses. At 28 dpi, both viruses induced comparable NA titers (Fig. 5i, Supplementary Fig. 13a). At 29 dpi, mice were challenged with wt ZIKV (strain Paraiba_01/2015). In contrast to the mock-immunized group, mice immunized with C/3′NCR-mir(T) or C/3′NCR-scr survived the challenge and did not develop detectable viremia (Supplementary Fig. 13b, 13c). Moreover, NA titers to ZIKV were not significantly boosted after the wt ZIKV challenge (Fig. 5i).

Hypersensitivity of AG129 mice to ZIKV infection can lead to overestimation of the C/3′NCR-mir(T) immunogenicity compared with that of an immunocompetent host. To resolve this concern, we evaluated the immunogenicity of the C/3′NCR-mir (T) in the rhesus macaques (Fig. 6a). All animals ($n = 4$) subcutaneously (s.c.) immunized with C/3′NCR-mir(T) developed detectable viremia, and the virus induced strong ZIKV-specific NA response, which by 56 dpi was comparable to the response induced by parental ZIKV-NS3m virus (Fig. 6b, c). Importantly, all immunized monkeys were protected from challenge with wt ZIKV (Fig. 6d). We concluded that cotargeting of ZIKV genome for a combination of CNS-, testis-, and epididymis-specific miRNAs does not interfere with induction of protective immunity in AG129 mice or an immunocompetent primate host.

## Discussion

To overcome the non-specific attenuating effect(s) of "foreign" sequence insertion on virus fitness, most viruses used in our study were designed in a way that requires only single modification of a variable region of the ZIKV 3′NCR. This resulted in only modest non-specific attenuation compared with a parental virus (Fig. 1). Insertion of miRNA targets into only one genomic region often results in their deletions under miRNA-mediated selected

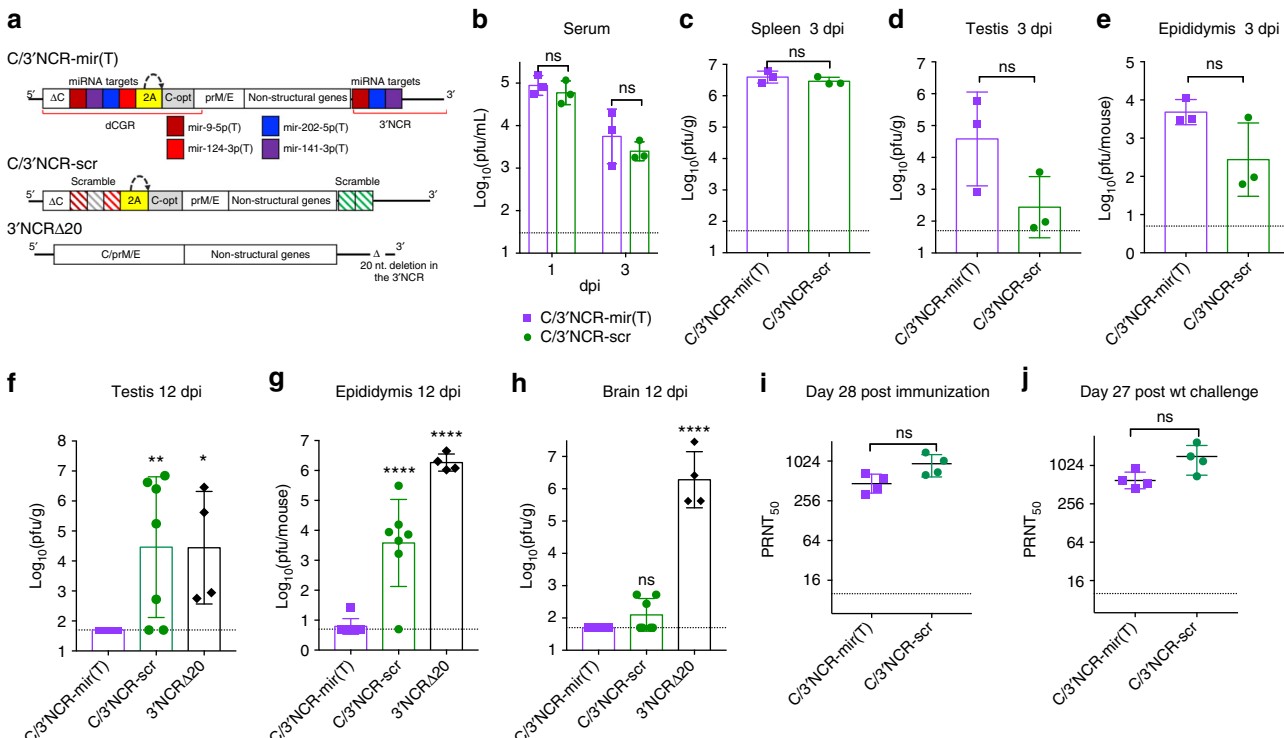

**Fig. 5** Replication and immunogenicity of C/3′NCR-mir(T) virus in adult AG129 mice. **a** Schematic representation of viral genomes used in the study. dCGR —duplicated capsid gene region; ΔC—truncated C gene; C-opt—a full-length copy of C gene containing synonymous mutations introduced in each AA codon (except AUG and UGG); colored boxes indicate miRNA targets for mir-9-5p (cherry), mir-141-3p (purple), mir-202–5p (blue), mir-124-3p (red); 2 A —autoprotease 2 A from foot-and-mouth disease virus (FMDV); the curved arrow indicates position of 2 A protease cleavage; striped cherry and red boxes indicate mutated targets for mir-9–5p and mir-124-3p, respectively; striped green box indicates scr sequence; striped gray box indicates random sequence of 21 nt. **b–h** mice were infected ip with $10^6$ pfu of the indicated viruses. Mean virus titer ± SD (shown as error bars) in the serum at 1 and 3 dpi **b**, spleen at 3 dpi **c**, testis at 3 dpi **d**, and 12 dpi **f**, epididymis at 3 dpi **e** and 12 dpi **g**, and brain at 12 dpi **h** was determined by titration in Vero cells. The dashed lines indicate the limit of virus detection. Differences between viral titers in the mouse serum or organs at 1 or 3 dpi were compared using unpaired two-tailed t test (ns denotes not statistically significant; $p > 0.05$). Differences between viral titers in the organs at 12 dpi were compared using one-way ANOVA (ns $p > 0.05$; *$p < 0.05$; **$p < 0.01$; ****$p < 0.0001$). **i–j** Mice were infected ip with $10^5$ pfu of C/3′NCR-mir(T) and C/3′NCR-scr. At 29 days post immunization, animals were challenged with $10^5$ pfu of wt ZIKV (strain Paraiba_01/2015). Neutralizing antibody titer in the serum of immunized mice at 28 dpi **i** and 56 dpi (27 days post challenge) **j** were compared using unpaired two-tailed t test (ns denotes not statistically significant ($p > 0.05$]). Horizontal lines denote geometric mean ± geometric SD (shown as error bars)

pressure[43–45], which can complicate the assessment of attenuation level of viruses with an intact miRNA target sequence in the specific organ(s). However, in our study escape mutant analysis was an important tool to precisely map ZIKV spreading routes within the MRT. It appears that the emergence of the escape mutants is a stochastic process that likely depends upon individual and temporal variations in the level of expression of a specific miRNA and the efficiency of interactions of the miRNA silencing machinery with targets inserted into the virus. Thus, suboptimal levels of expression of specific miRNA in a given cell and a prolonged access of the virus to that cell can favor the emergence of escape mutants.

Similar to infection with natural ZIKV isolates[19], the 2×scr virus replication in the mouse testis appeared to be a biphasic process (Fig. 1f)[17,18,46,47]. Testicular infection with the 2×scr virus during early viremic phase was initiated within the interstitial compartment, which contains blood and lymphatic vessels, implicating the hematogenous/lymphogenous virus spread. The second phase of infection coincided with extensive 2×scr replication in virtually all types of cells comprising the seminiferous tubules (Fig. 3e). This suggests that, during spread from the interstitium, the virus switched host cells and began to efficiently replicate in accessible Sertoli cells and spermatogonia with subsequent infection of other germ cells. In general, this pattern of

ZIKV spread in the testis is similar to that reported by other investigators[17–20,46,47], indicating that insertions of heterologous sequence into the 3′NCR of ZIKV did not have a substantial impact on its pathogenicity.

The replication pattern of the 2×scr virus in the mouse epididymis in many aspects mirrored that seen in the testis (Fig. 1–4). In both organs, virus replication appeared to be a biphasic process. The first phase coincided with virus infection of the interstitial compartment (Fig. 4e), whereas the second phase was associated with infection of the epididymal epithelium (Fig. 4h). This is also consistent with the pattern of infection by natural ZIKV isolates[18–20]. A biphasic profile of 2×scr replication in both organs of the murine MRT suggests that the virus encountered a structural tissue barrier (such as the peritubular myoid cells and/or basal lamina) which slowed down its dissemination, and/or switched the host cells to replicate more efficiently. This may explain a high variability of 2×scr titers during transition between the two phases of replication (6–9 dpi for testis, 6 dpi for epididymis).

We showed that insertion of targets for testis-specific mir-202–5p (and mir-465a-3p) selectively inhibited the late phase of ZIKV replication in the testis, when the virus gained access to the cells residing within the seminiferous tubules, but not the first phase when the virus was seeding the testicular interstitium by

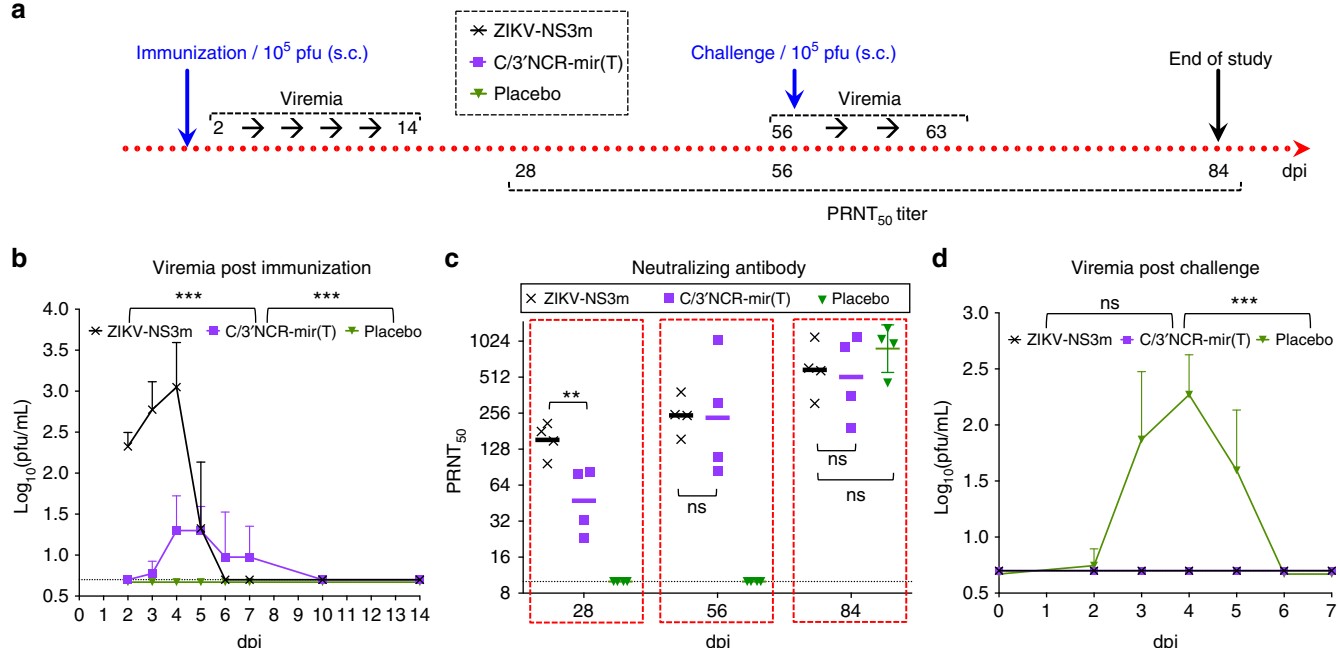

**Fig. 6** C/3′NCR-mir(T) virus is immunogenic to the rhesus macaques. **a** Experimental design of the study. **b** Mean viremia ± SD (shown as upper error bars) post immunization with ZIKV-NS3m or C/3′NCR-mir(T) viruses. **c** Neutralizing antibody response in immunized monkeys before and after the challenge with $10^5$ pfu of wt ZIKV. Horizontal lines denote geometric mean ± geometric SD (shown as error bars). **d** Mean viremia + SD (shown as upper error bars) after the challenge with wt ZIKV. Differences between viral titer and between ZIKV-specific NA titer in serum were compared using two-way ANOVA and one-way ANOVA, respectively (**$p < 0.01$, ***$p < 0.001$; ns denotes not statistically significant ($p > 0.05$). The dashed lines indicate the limit of virus detection (0.7 $\log_{10}$(pfu/mL)) for serum **b** and **d**, or the limit of NA titer detection **c**

hematogenous/lymphogenous route (Fig. 2, 3, Supplementary Fig. 10). These observations are consistent with the abundant expression of mir-202–5p in the cells comprising the seminiferous tubules, including Sertoli cells, spermatogonia, spermatocytes, and spermatids (Supplementary Fig. 4)[32,48,49], but not in the cells residing in the interstitium (including those of a hematopoietic origin[33]). Despite relatively high level of expression in the testis, targeting of ZIKV for mir-449a-5p did not affect accumulation of 2×449a(T) in this organ. Analysis of published data indicated that, compared to other testis-expressed miRNAs used in our study, mir-449a-5p is the least expressed in spermatogonia (but not in spermatocytes and spermatids) (Supplementary Fig. 14)[32]. In addition, mir-449a-5p is not expressed in the Sertoli cells[50], whereas both mir-202–5p and mir-465a-3p are highly expressed in this cell type[49]. Therefore, the fact that Sertoli cells and spermatogonia are now established as the main targets of ZIKV in the testis[17,18,20], but these cells do not express (or express at very low levels) mir-449a-5p, may explain the failure of mir-449a-5p targeting to inhibit ZIKV replication in the testis.

Analysis of the replication of viruses targeted for the testis-expressed mir-202–5p and mir-465a-3p (2×202(T) and 2×465a(T), respectively) demonstrated that ZIKV infection of the testis is not a prerequisite of infection of the "downstream" organ, the epididymis. If we assume that ZIKV infects the epididymis exclusively via the excurrent ducts from infected testis, then the inhibition of ZIKV replication in the testis should have slowed down or prevented the infection of the epididymal epithelium. However, reduced replication of the Te-2×202(T)-stb virus in the testis was not associated with a comparable reduction of Te-2×202(T)-stb titer in the mouse epididymis (Fig. 2, 4). Moreover, deletions that were detected in the Te-2×202(T)-mut virus isolated from the testis were not detected in the virus isolated from epididymis of the same mice at 12 dpi (Supplementary Table 1). These findings strongly suggest that ZIKV can disseminate to the

epididymis by another route, most likely hematogenous/lymphogenous, followed by replication in the interstitium (Fig. 4f) and subsequent infection of the epididymal epithelium, a cell type that was predominantly infected at 12 dpi (Fig. 4j).

To inhibit a hematogenous route of ZIKV dissemination we selected mir-141–3p, which is specifically expressed in epididymal epithelium. Despite its inability to infect epididymal epithelium, the 2×141(T) attained a titer in the epididymis similar to 2×scr. This suggests that ZIKV can be transported to this organ from the testis, which could occur simultaneously with the hematogenous/lymphogenous dissemination. Therefore, we reasoned that: (1) a simultaneous co-targeting for mir-202–5p and mir-141–3p should result in a significant reduction of ZIKV load in the epididymis; (2) if the epididymal infection is restricted by the presence of targets for mir-141–3p, then escape mutations in mir-202–5p targets, which sporadically occur in the testis, should be detected in the epididymis. Both these predictions were corroborated experimentally using viruses simultaneously expressing targets for mir-202–5p and mir-141–3p (Fig. 2–5, Supplementary Table 3). Moreover, using various target combinations for different testis- and epididymis- specific miRNAs, we showed that attenuation of the 2×202/141(T) in the epididymis was not an unique event, but was a common consequence of simultaneous restrictions of both hematogenous/lymphogenous and excurrent testicular routes of infection by dual miRNA targeting (Supplementary Fig. 10).

Restriction of individual dissemination routes for viruses 2×202(T) (testicular) or 2×141(T) (hematogenous) suggests, that compared with 2×scr, there should have been some reduction in the titer of each of these viruses in epididymis. However, this was not detected in our experiments. We speculate that variation between viral titer in the epididymis of different mice at 12 dpi was substantially greater compared with relatively small differences (twofolds), which are expected owing to a restriction of

each of the two dissemination routes. Also, considering that ZIKV reaches maximum titer in the epididymis slightly faster than in testis, we think that restriction of hematogenous pathway by mir-141–3p targeting would have more pronounced effect on attenuation of the 2×141(T) virus in the epididymis, if it was assessed during earlier times post infection (6 or 9 dpi) compared with that observed at 12 dpi.

In the context of vaccine research, evaluation of the "worst case scenario" outcomes of ZIKV infection using immunodeficient AG129 model can ensure vaccine safety for all recipients, whose immunological status could be unknown. However, the reliance on this model to study pathogenesis of ZIKV could generate results that might not be fully relevant for an immunocompetent host whose cells might be less permissive to infection. Obviously, ZIKV infection and dissemination in human testis and epididymis remains to be fully elucidated. It would be, therefore, important to evaluate ZIKV dissemination routes in the MRT of non-human primates (NHPs) or in the MRT of recently developed immunocompetent hSTAT2 KI mice using mouse adapted strain of ZIKV[51]. Interestingly, viral dissemination between testis and epididymis of NHPs was demonstrated for another sexually transmitted pathogen—simian immunodeficiency virus[52], suggesting that this route of spreading could be utilized by ZIKV in primates as well. ZIKV has also been detected in the semen of vasectomized men[53,54] and mice[19] implicating vas deferens, seminal vesicles, and the prostate as important organs potentially contributing to ZIKV shedding into semen. Future work should be focused on understanding of a relative contribution of the individual MRT organs to the ZIKV infectivity, persistence, and sexual transmission. The miRNA-targeting approach for restriction of ZIKV replication in each component of the MRT individually, and viruses with altered tissue tropisms described here, represent valuable experimental tools to address these and other outstanding questions regarding ZIKV infection of the MRT[47].

The major concern of the miRNAs co-targeting approach for live virus vaccine development continues to be the emergence of escape mutants that are resistant to miRNA-mediated inhibition and the potential for virus reversion to a virulent phenotype. Viruses carrying target sequences only in the 3′NCR (2×202(T), 2×141(T), 2×202/141(T)) were capable of selecting deletions within the miRNA targets during replication in the MRT. The most effective ZIKV suppression in the MRT was achieved when two target cassettes for testis- and epididymis-specific miRNAs (mir-141–3p and mir-202–5p) were inserted into two genome regions (dCGR and 3′NCR) of the C/3′NCR-mir(T) virus (Fig. 5). This simultaneous miRNA co-targeting of ZIKV genome resulted in improved genetic stability and restricted virus replication in both, the testis and epididymis. In contrast, the deletion-based strategy to attenuate ZIKV, which has been widely used for the development of other live-attenuated flavivirus vaccines[41,42,55], did not have an effect on specific ZIKV tissue tropism. The 3′NCRΔ20 virus was still able to replicate to high titers in the brain, testis, and epididymis of AG129 mice (Fig. 5f–h). Most importantly, the miRNA-targeted C/3′NCR-mir(T) vaccine candidate virus retained the ability to replicate in serum and peripheral tissues of mice and rhesus monkeys during the early course of infection (Fig. 5, 6) and induced a strong NA response against ZIKV, which was sufficient to completely protect the immunized animals against challenge with wt ZIKV virus (Fig. 5, 6, Supplementary Fig. 13).

Overall, the C/3′NCR-mir(T) virus is attenuated for specific tissues, yet immunogenic, and represents a promising genetic platform for further development of live-attenuated ZIKV vaccine candidates. Future studies are needed to identify miRNA targets to restrict ZIKV tropism in the maternal/fetal tissues. Given a high level of mir-141–3p expression in the trophoblasts[28,56], we anticipate that ZIKV genome co-targeting for this miRNA may be a crucial component of ZIKV attenuation not only for the epididymis, but also for human placenta.

## Methods

**Statement of compliance**. All experimental protocols were approved by the NIH Institutional Biosafety Committee. All animal study protocols were approved by the NIAID/NIH Institutional Animal Care and Use Committee (IACUC) and performed in compliance with the guidelines of the NIAID/NIH IACUC. The NIAID DIR Animal Care and Use Program acknowledges and accepts responsibility for the care and use of animals involved in activities covered by the NIH IRP's PHS Assurance D16–00602 (formerly A4149–01) that was last approved 6/30/2015.

**Plasmids**. Annotated sequences for all plasmids used in the study are available from the authors upon request. Vero-cell-adapted cDNA clone (ZIKV-NS3m) of ZIKV[22] was constructed based on the strain Paraiba_01/2015, isolated during the 2015 epidemic in Brazil. To generate 2×scr, a 20 nt sequence (scr) of the eGFP coding sequence (nts: 241–260)[57] was introduced at nt positions 8 and 14 of the 3′ NCR as depicted in Supplementary Fig. 1. This sequence is located in the variable region of the 3′NCR of flaviviruses, where insertion of short heterologous sequences typically does not cause a substantial reduction of viral replication fitness[43,44]. The scr sequence has no matching sequences in mouse or human genomes. A panel of miRNA-targeted ZIKV clones was produced by replacing the scr sequence in the 2×scr clone with complementary sequences (targets) for human miRNAs: mir-9–5p, mir-124–3p, mir-141–3p, mir-202–5p, mir-449a-5p (Supplementary Table 1), mir-465a-3p or mir-200c-3p (Supplementary Fig. 10c) generating constructs 2×9(T), 2×124(T), 2×141(T), 2×202(T), 2×449a(T), 2×465a(T) and 2×200c(T), respectively (Supplementary Fig. 1, 10c). As a unique site for restriction endonuclease NsiI, which was used for 2×scr cloning, is present in a target sequence of mir-202–5p, it was replaced with an XhoI site. Although all inserted miRNA targets were designed based on sequences of human miRNA, they are also complementary to corresponding mouse miRNAs. To generate 2×202/141 (T) (Supplementary Fig. 1), two copies of mir-202–5p and mir-141–3p targets were inserted between nts 7 and 14 in the 3′NCR of ZIKV-NS3m (Supplementary Fig. 1). To generate 2×465a/141(T), each target for mir-202–5p in the 2×202/141 (T) was replaced with targets for mir-465a-3p (Supplementary Fig. 10c). To generate 2×202/200c(T), each target for epididymis-specific mir-141–3p in the 2×202/141(T) was replaced with targets for mir-200c-3p (Supplementary Fig. 10c).

To generate C/3′NCR-mir(T), we first modified ZIKV-NS3m by duplicating its C gene region (Supplementary Fig. 11) using a strategy that was reported earlier[40]. An open-reading frame (ORF) shifting mutation was introduced by inserting a single adenine residue after 11 nt of coding C gene sequence. This modification was performed upstream of the ZIKV cyclization sequence and was designed to exert a minimal effect on the secondary RNA structure, preserving the functionality of cis-acting elements of the C gene. A cassette containing target sequences for mir-9–5p, mir-141–3p, mir-202–5p, and mir-124–3p was inserted after codon 50 of the C gene (C-trn) preserving the uninterrupted ORF. A sequence of 2 A protease from the foot-and-mouth disease virus was fused with a full-length copy of the C gene and inserted downstream of mir-124-3p target in the same reading frame as the first AUG codon at the 5′-end of ZIKV ORF. To prevent homologous recombination between C-trn and full-length copy C gene, synonymous mutations were introduced in each codon (except AUG and UGG) of the full-length C gene sequence (designated C-opt) impairing cis-acting elements (including the cyclization sequence), but preserving the amino acid sequence of C protein. The ZIKV-NS3m with a dCGR was subsequently modified by introducing target sequences for mir-9–5p, mir-202–5p, and mir-141–3p into the 3′NCR of ZIKV between nts 8 and 14, generating C/3′NCR-mir(T) (Supplementary Fig. 11).

To generate C/3′NCR-scr (Supplementary Fig. 12), we modified the dCGR of C/3′NCR-mir(T) by introducing synonymous substitutions in every codon in the sequences encoding mir-9–5p and mir-124–3p target, followed by replacement of mir-202–5p and mir-141–3p targets with a random 21 nt sequence. The resulting dCGR was used to substitute for the original C gene in 2×scr clone.

To generate 3′NCRΔ20, we deleted 20 nt sequence (positions: 260–280 nt) from the 3′NCR of ZIKV-NS3m as originally described by Shan et al.[41,55].

**DNA transfection, virus recovery, and titration**. Vero (African green monkey kidney) cells were obtained from the World Health Organization and were used between passages 141 and 149. Cells were propagated at 37 °C and 5% $CO_2$ in Opti-Pro medium (Gibco) supplemented with 4 mM L-glutamine[58]. All recombinant viruses were recovered by transfection of 2.5 μg plasmid DNA of the respective infectious clone into $1.5 \times 10^6$ Vero cells seeded in a 12.5-cm[2] flask in complete Dulbecco's Modified Eagle's medium (DMEM) supplemented with 10% fetal bovine serum (FBS) and 1× penicillin–streptomycin–glutamine solution (Invitrogen)[22]. Three days after DNA transfection, Vero cell supernatants were clarified by 5 min (min) centrifugation at 3000×g, followed by the addition of 1× SPG (218 mM sucrose, 6 mM L-glutamic acid, 3.8 mM $KH_2PO_4$, 7.2 mM $K_2HPO_4$, pH 7.2)[59], aliquoted, and stored at − 80 °C. Recombinant viruses harvested after DNA transfection were used in all experiments without additional propagation.

To compare growth kinetics of miRNA-targeted viruses in cell culture, Vero cells in 12.5-cm$^2$ flasks were transfected in duplicates with 2.5 μg plasmid DNA as described above. Transfected cells were maintained in 5 mL of complete DMEM at 37 °C and 5% CO$_2$. Aliquots of cell culture medium (0.5 mL) were collected daily and the volume of cell culture supernatant was restored by adding 0.5 mL of complete DMEM. Aliquots of cell culture medium were stored at − 80 °C and titrated by plaque assay in Vero cells. Differences in virus replication kinetics were compared using two-way ANOVA analysis implemented in Prism 7 software (La Jolla, CA).

Virus titers were determined by plaque assay in Vero cells seeded in 24-well plates and overlaid with 1 mL/well of Opti-MEM (Gibco) containing 1% methylcellulose, 2% FBS, 2 mM L-glutamine, and 50 μg/mL of gentamicin[22]. Plates were incubated at 37 °C and 5% CO$_2$ for 4–5 days. Cells were washed three times with PBS, fixed for 20 min with 100% methanol, and stained with 0.5% crystal violet solution.

**Genetic stability of C/3′NCR-mir(T) in Vero cells.** For the first passage, a confluent monolayer of Vero cells in a 25-cm$^2$ flask were infected with C/3′NCR-mir(T) virus at a multiplicity of infection of 0.01. Cells were maintained in 5 mL of Opti-Pro medium supplemented with 4 mM L-glutamine and 2% FBS at 37 °C and 5% CO$_2$. At 3 or 4 dpi, the supernatant was harvested and diluted 1/50 with Opti-Pro medium. One milliliter of diluted virus was used to infect 25-cm$^2$ flasks of fresh Vero cells. The process was repeated nine times. Genetic stability of the virus was evaluated at the end of the 10th passage. Cell supernatant was clarified by 5 min centrifugation at 3000×$g$, and RNA was extracted using the QIAamp Viral RNA Mini kit (Qiagen). Viral RNA was PCR-amplified using the Transcriptor One-Step RT-PCR Kit (Roche) and two pairs of ZIKV-specific primers: (1) ZV-1-F and ZV-451-R, which flank the region of the miRNA target insertion in the dCGR; or (2) ZV-10044-F and ZV-10722-R, which flank the region of the miRNA target insertion in the 3′NCR (Supplementary Table 6). Amplicons were purified by agarose gel electrophoresis and sequenced using Sanger sequencing and primers ZV-451-R or ZV-10722-R.

**Replication of miRNA-targeted ZIKVs in adult AG129 mice.** An AG129 mouse colony was established from a breeding pair purchased from Marshall BioSources. The colony was maintained at the NIAID/NIH animal facility. Recombinant ZIKVs were diluted in L-15 media (Gibco) supplemented with 1× SPG solution (L-15/SGP) to a concentration of 10$^7$ pfu/mL. Adult (4–6-week-old) male AG129 mice were injected ip with 10$^6$ pfu of each virus (0.1 mL/mouse). At various time intervals post infection, mice were bled and euthanized, followed by harvesting the organs of interest. Mouse organs and serum samples designated for determination of virus load were stored at − 80 °C. Organs used for histological examination were immediately placed in 5 mL of Perfusion Fixative Super Reagent (Electron Microscopy Sciences) for fixation. To evaluate neurovirulence of selected viruses, infected mice were monitored daily for onset of neurologic disease (paralysis/lethargy) for 28 dpi.

**Immunogenicity of miRNA-targeted ZIKVs in adult AG129 mice.** For mouse immunogenicity studies we used a lower dose of inoculation (10$^5$ pfu/mouse) as compared with the infection dose, which was used in the ZIKV pathogenesis studies (10$^6$ pfu/mouse). The 10$^5$ pfu dose was selected to be consistent with the dose, which is used in the monkey immunogenicity study (see below). Adult male AG129 mice were injected ip with 0.1 mL of L-15/SGP (MOCK- control), or with C/3′NCR-mir(T) or C/3′NCR-scr viruses diluted in L-15/SGP to 10$^6$ pfu/mL (dose —10$^5$ pfu/mouse). At 28 dpi, mice were bled to determine NA titers. At 29 dpi, mock- or virus-inoculated mice were challenged with 10$^5$ pfu of wt ZIKV (Paraiba_01/2015) and were monitored for onset of neurologic disease (paralysis) for 27 days. Mice were bled at 2 and 27 days post challenge (dpc) to determine wt ZIKV titers in the serum and NA titer, respectively.

**Evaluation of virus titer in mouse organs and serum.** At various intervals post infection, the brain, spleen, pair of testicles, pair of epididymides and 0.05–0.1 mL of serum were collected from each mouse. Brain, spleen, and testes were weighed, and 10% homogenates were prepared in L-15/1× SPG solution. Virus titers in each homogenate were determined by a plaque assay in Vero cells[22,58]. Epididymis is engulfed in a fat tissue, which remains attached to epididymides during dissection. Owing to the difficulties of controlling the amount of fat tissue, which is picked up during epididymis dissection, we reasoned that normalization of viral titers to epididymis weight might be misleading. Therefore, a pair of epididymides from each mouse was homogenized in 1 mL of L-15/1× SPG solution, and viral titer was assessed in Vero cells and normalized to pfu per mouse. Serum samples were diluted in five volumes of L-15/1× SPG solution (6× dilution factor) and were titrated by plaque assay in Vero cells.

**Stability of miRNA-targeted viruses in mouse organs and serum.** Aliquots (0.14 mL) of a homogenized mouse organs or sixfold-diluted serum samples were also taken for RNA extraction using the QIAamp Viral RNA Mini kit (Qiagen). Viral RNA was PCR-amplified using the Transcriptor One-Step RT-PCR Kit (Roche) and a pair of ZIKV-specific primes: ZV-10044-F and ZV-10722-R, which

flank region of the miRNA target insertion in the 3′NCR (Supplementary Table 6). Amplicons were purified by agarose electrophoresis and sequenced using primer ZV-10722-R.

**Evaluation of NA titer in mouse and monkey serum.** NA titer in mouse serum was determined using the 50% plaque reduction neutralization assay (PRNT$_{50}$)[60] against ZIKV-NS3m virus[22]. In brief, mouse serum was 10-fold diluted with complete Opti-Pro medium and heated for 30 min at 56 °C. The heated-inactivated serum was fourfold diluted with a complete Opti-Pro medium and 0.12 mL of diluted serum was mixed with 0.12 mL of ZIKV-NS3m virus at concentration of 800 pfu/mL. The virus/serum mix was incubated for 30 min at 37 °C, and 0.1 mL of virus/serum solution was added to the duplicate wells of Vero cells seeded in the 24-well plate. Cell were incubated at 37 °C for 1 h, followed by washing with 1 mL of complete Opti-Pro medium. Cells were overlaid with 1 mL/well of Opti-MEM containing 1% methylcellulose, 2% FBS, 2 mM L-glutamine, and 50 μg/mL of gentamicin. Plates were incubated at 37 °C and 5% CO$_2$ for 3 days, fixed with 100% methanol, and stained with 0.5% crystal violet solution.

NA titer in monkey serum was determined as described above with minor modifications: Opti-MEM containing 2% FBS, 50 μg/mL gentamicin and 0.5% albumin was used for monkey serum dilutions; wt ZIKV (Paraiba_01/2015) was used at concentration of 600 pfu/mL; Vero cells were overlaid with methylcellulose/Opti-MEM without washing; cells were fixed with 80% methanol; plates were immunostained using mouse anti-flavivirus envelope protein antibody 4G2 (produced in house; the dilution factor is 1:2000).

**Immunohistochemistry and microscopy.** Tissues (testes and epididymis) from ZIKV-infected (and uninfected control) AG129 mice were fixed in 4% paraformaldehyde prior to embedding and processing in histological grade paraffin. Each paraffin block was designed to include the test tissue samples together with positive (2×scr virus-infected) and negative (mock) controls. Blocks were sectioned at 5 μm and stained with hematoxylin–eosin for examination by light microscopy. Immunohistochemical staining was performed on a Leica Bond-RX automated system according to manufacturer recommended protocol. Tissue sections were heated to 72 °C for 30 min in Bond Dewax Solution (Leica) then rehydrated with absolute alcohol washes and 1× ImmunoWash (ACR-024, StatLab). After that sections were heated to 100 °C for 20 min in Bond Epitope Retrieval Solution 1 (Leica) for heat-induced epitope retrieval. After exposure to peroxide block (Leica) for 5 min, tissues were incubated with the anti-Zika virus NS2B antibody (1:500 dilution factor; GTX133308, GeneTex). Following incubation with primary antibody, the tissue sections were rinsed with 1× ImmunoWash and viral antigen was detected with 3,3′-diaminobenzidine chromogen using the Bond Polymer Refine Detection Kit (DS9800, Leica). Counterstaining was achieved with hematoxylin. Whole tissue section imaging was performed at ×40 magnification using the ScanScope AT2 (Leica Biosystems). Aperio eSlide Manager and ImageScope software were used for digital slide organization, viewing, acquisition, and analysis.

**Quantification of miRNA expression in mouse organs.** Snap-frozen tissues extracted from three male AG129 mice at 7 weeks of age were pulverized with a series of sharp blows delivered with a 2-pound hammer. A portion of the pulverized tissue (< 100 mg) was homogenized once for 20 s in lysing matrix D tubes (MP Biomedicals, Santa Ana, CA) containing 1000 μl Trizol (Thermofisher Scientific, Waltham, MA) in a FastPrep® FP 120 instrument (MP Biomedicals) at a speed of 6.0 meters per second. Homogenized Trizol lysate was extracted as described before[61]. The small RNA yield was determined by spectrophotometry at 260 nm and 280 nm. Small RNA size was assessed using the Agilent 2100 Bioanalyzer using small RNA kit (Agilent Technologies, Santa Clara, CA). miRNA yield varied from 0.6 to 2.2 μg.

miRNA libraries for next-generation sequencing (NGS) were prepared with 100 ng of purified small RNA using the TruSeq Small RNA Library Prep Kit (Illumina, San Diego, CA) following the manufacturer's guidelines without modification. Libraries were quantified with the Kapa Quantification Kit for Illumina Sequencing (Roche, Indianapolis, IN) and pooled equitably for single-read 50 cycle sequencing on the HiSeq 2500 (Illumina, San Diego, CA).

NGS resulted in an average of 24.7 million reads per library. Raw reads were trimmed of Illumina adaptor sequence using cutadapt version 1.12[62] followed by filtering for low quality reads using fastq quality filter from the FASTX-Toolkit[63]. Remaining reads were aligned to the *Mus musculus* ribosomal DNA, complete repeating unit (BK000964.1) using Bowtie2[64]. Reads that did not align to ribosomal DNA were aligned to a database of 1978 *Mus musculus* mature miRNAs from miRBase v22[65]. Reads aligning to mature miRNA were counted with custom scripts, followed by normalization and differential expression analysis using EdgeR[66]. To construct Fig. 4a the normalized read count of value "0" for mir-202–5p in the epididymis of one of the three mice was replaced with the smallest positive value (1.09) for this miRNA in epididymis of the other two mice used in the study.

**Validation of the miRNA expression by in situ hybridization.** FFPE sections (5 microns; two per slide) were mounted on FisherBrand Plus slides and baked at 60 °C for 30 min. Tissues were deparaffinized for 5 min in xylene, immersed in 100%

ethanol for 5 min then air-dried. LNA probes mmu-miR-141–3p, mmu-miR-202–5p, and Scramble-miR, were prepared according to the manufacturer's recommended conditions (Exiqon/Qiagen) and each was labeled at the 5' end with digoxigenin. In situ hybridization (ISH) optimization for miR-141–3p and miR-202–5p detection was done according to a previously published protocol[67]. Optimal conditions for ISH were antigen retrieval for 30 min and 30 pmoles/section (3 pmol/μL) of probe in Enzo hybridization buffer (ENZ-33808). Probe solution was placed on tissue sections, covered with polypropylene coverslips and heated to 60 °C for 5 min, followed by hybridization at 37 °C overnight. Sections were washed in intermediate stringency solution (0.2× SSC with 2% bovine serum albumin) at 55 °C for 10 min. Sections were treated with anti-digoxigenin—alkaline phosphatase conjugate (1:150 dilution in pH 7 Tris buffer; Roche) at 37 °C for 30 min. Development was carried out with NBT/BCIP from ThermoFisher (34042). Development was closely monitored and stopped when the control sections appeared light blue. Development time with the chromogen was between 15–30 min. Sections were counterstained with nuclear fast red for 3–5 min, rinsed, and mounted with coverslips.

**Study of ZIKV immunogenicity in non-human primates**. Viruses were evaluated for replication and immunogenicity in rhesus macaques (Macaca mulatta) shown to be seronegative for ZIKV, dengue virus, and West Nile virus. On study day 0, monkeys were injected s.c. in the upper shoulder area with $10^5$ pfu of virus (1 mL) diluted in 1× L-15 medium (Lonza) or with a placebo inoculum of diluent. Blood was collected on study days 0, 2–7, 10, and 14 for viremia determination and on days 28 and 56 for immunogenicity (NA titer), processed for serum, and stored at −80 °C. Viremia was determined for each day by direct plaque assay in Vero cells, and serum NA titer was determined for days 0, 28, and 56 by $PRNT_{50}$ on Vero cells using wt ZIKV (Paraiba_01/2015) as target. On day 56 post inoculation, all monkeys were challenged by subcutaneous infection with $10^5$ pfu of wt ZIKV (Paraiba_01/2015). Blood was collected on study days 56–63, and 84, processed for serum, and stored at −80 °C. Viremia and NA responses were determined as described above.

## Data availability

The authors declare that the data supporting the findings of this study are available with the article and its Supplementary Information files, or are available from the authors upon request.

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

## Acknowledgements

We thank Evgeniya Volkova for her comments and suggestions during manuscript preparation. We thank Bryant Maldonado, Stacy Ricklefs, and Kimmo Virtaneva for technical assistance. We thank Dr. Gerard Nuovo, Jim Williams, and Dr. Adel Mikhail (Phylogeny) for their excellent technical support with detection of miRNAs by in situ hybridization. This work was supported by the Division of Intramural Research Program of the National Institute of Allergy and Infectious Diseases, National Institutes of Health.

## Author contributions

K.A.T., O.A.M., S.S.W., and A.G.P. designed the experiments; K.A.T., O.A.M., G.L., H.K., N.T., J.M.G., L.M., B.M.N., I.M., C.M., E.AC., E.W.L., and A.G.P. performed the research; K.A.T., O.A.M., I.M., S.S.W. C.M., and A.G.P analyzed the data; K.A.T., O.A.M., M.E.B., I.M., S.S.W. C.M., and A.G.P wrote the paper. All authors reviewed the final draft of the manuscript.

## Additional information

**Competing interests:** The authors declare no competing interests.

