## [Peer Review File · Nature Communications]

Reviewers' Comments:

Reviewer #1:

Remarks to the Author:

The manuscript by Tsetsarkin et al. is well-written and describes a novel attenuated zika vaccine that has reduced pathogenicity in a sensitive mouse model. Targeted sequences introduced into a molecular clone allow reduced replication in the brain and testes, important replication sites that could have consequences for severe disease or transmission in a vaccination setting. This is an interesting strategy and this work represents a novel approach for improved safety for vaccination. While I would recommend this work for publication in Nature Communications, there are some issues that should be addressed prior to acceptance of the manuscript.

Major issues

- 1- In figure 1, the virus titer curve for epididymis is truncated and does not follow the same time-course as the other tissues. Various assumptions are made later in the manuscript that are based on limited observations of virus titer in the epididymis. It would be helpful if this tissue were evaluated at the same times as the other tissues. It is assumed that the tissue titer declines in the epididymis as it does in the testis, but this may not be the case and should be evaluated and included in the curve.
- 2- It would be useful to show virus replication curves generated in cell culture in order to compare wt and modified ZIKVs in order to better compare the replication of these viruses over time.
- 3- Could an explanation be given to why mir-449a-5p may not have worked to suppress viral growth in the testis? I do think it is useful to include negative data in the manuscript, but there should be more information included. The information is so brief on this construct that it is questionable why it is included at all.

Minor issues:

- 1- Page 6, line 96. Please include a discussion of the figures and subfigures in order. I.e, the first subfigure mentioned is 1G.
- 2- Page 7, line 108-109. This figure is not specific to the testis, so please modify this section name. I would also suggest moving the current figure 2F to the 2D position and 2D to 2E and 2E to 2F, so the figure is in order in relation to time.
- 3- Page 10, line 161. In addition to the testis, there is also significant reduction in the brain titers. Please include some indication why this might be the case.
- 4- Page 18, line 270. Please include these abbreviations earlier in the manuscript where they first occur.
- 5- Page 20, line 301, suggested change to "...from Te-2x202/141(T)-stb-infected mice..."
- 6- Page 26, line 403 and throughout. I prefer not to see references to the figures in the discussion and suggest removing these.
- 7- Page 35, lines 593-594, 605-606. Many different virus challenge doses were used. Please provide justification why these weren't kept consistent.

Reviewer #2:

Remarks to the Author:

This is a very well designed study by investigators with expertise in vaccine studies. The authors use the approach of targeting miRNA to achieve selective restriction of ZIKV replication in the testis and epididymis. They show that a combined co-targeting of ZIKV genome for the CNS-, testis- and epididymis-specific miRNAs restricts ZIKV infection of these organs, but does not impair the development of humoral immunity to ZIKV infection in mice. Strengths of the study include very strong data and innovative experimental approach to investigate ZIKV trafficking in the epididymis that can also be utilized for other models and other viruses. However, there are major concerns that the authors did not address.

- The use of AG129 mice is a significant limitation of the study and should be addressed. Being deficient in both type I and type II IFN receptors makes it a very artificial model and severely diminishes the predictive capacity of their results. IFNAR in particular allows uninfected cells to be alerted and mount an ISG response before coming in contact with the virus. ZIKV NS5 targets STAT2 for degradation in infected cells only, thus intact IFNAR is still important for dissemination/trafficking. Its very unlikely that the trafficking of the virus would occur the same way within the same cells in an immunocompetent model, regardless of STAT2 antagonism by ZIKV NS5.
- Use of AG129 mice limits the predictive capacity of their approach in humans and nonhuman primates – in an immunocompetent scenario, many cells in the MRT may not be readily permissible to ZIKV infection, and therefore trafficking of virus through MRT would be different. Interpretation of ZIKV trafficking through the MRT may only be relevant to AG129 mice. These issues should be addressed by validating some of the experiments in immunocompetent mice treated with antibody to IFNs during infection to facilitate first wave of infection and tissue invasion of the virus. Addressing these limitations would strengthen the manuscript overall.
- Discussion: Although their data showing protection of C/3'NCR-scr immunized AG129 mice to wt ZIKV challenge is compelling, however considering the limitation of the animal model, do the authors expect this type of protection to occur in other models? Authors should include their thoughts in the discussion.
- At many places, the authors should clarify the statement by including either 'AG129' or 'immunocompromised mice'. For instance, line 368 should probably state "protective immunity in AG129 mice". Also the title should include '... virus dissemination in immunocompromised mice'
- In some experiments comparison of 2xscr replication with wild-type virus would be helpful- specially replication in the testes and epididymis?

Reviewer #3:

Remarks to the Author:

Tsetsarkin et al. used miRNA target sites insertion in ZIKV genome to restrict its replication in selected organs and elucidate the routes of ZIKV infection in the male reproductive tract. Insertion of mir-202 (testis specific) and mir-141 (epididymis specific) target sites prevented infection of seminiferous tubules in testis and epithelial cells in epididymis respectively. They also observed that epididymis infection occurred via two routes, blood-associated and testicular. Finally, they generated a virus with restricted replication in CNS, testis and epididymis but a full humoral immunity potential. This study is interesting and brings convincing results for specific restriction of virus replication using miRNAs and promising data for a potential use for the development of a safe live-attenuated ZIKV vaccine. However, the part concerning the dual-route infection of epididymis is very weak and should be improved.

Major points:

1) Authors should add a minimum of miRNA quantification data and not depend fully on published data knowing that miRNA expression level can display important differences between individuals. It would be good to have at least mir-202 quantified in testis samples from the 12 mice corresponding to 2x202(T)-stb and 2x202(T)-mut. This could bring interesting details on: 1) mir-202 level variation between individuals. 2) Is there a negative correlation between miR-202 and virus levels in stb samples? 3) Comparing mir-202 levels in stb and mut samples could indicate whether escape mutants are associated with higher or lower level of selective pressure. At the very least, if frozen samples are no more available for miRNA quantification, the authors should discuss the parameters affecting the level of miRNA mediated selection pressure and/or emergence of escape mutants.

2) Figure 3 and Figure 4: characterization of interstitial labeling at day 12pi is missing in order to close the story of biphasic course of infection at the immunohistochemistry level. The authors should add pictures and quantitative analysis of interstitial labeling (immunohistochemistry) for testis and epididymis at day 12 pi.

3) Seminiferous tubules are composed of germ cells and somatic Sertoli cells, 2 very distinct cell lineages with little in common. Therefore it would be important to precise whether mir202 is expressed in Sertoli or germ cells as it is unclear from the quoted references with discrepant results. This would be most useful for further approaches using mir202 (eg for viruses with different cell tropism) and may also shed light on the sequence of infection in the seminiferous tubules (eg Sertoli or germ cell mediated). For instance ISH on whole tissue or PCR on isolated cells could be performed.

4) The weakest point of the paper is from far the demonstration of the dual-route (blood and testis) of epididymis infection. Figure 4A shows that 2x141(T) virus level in epididymis at 12 dpi is the same as 2x202(T); however, immunohistochemical analysis clearly demonstrates that this corresponds in fact to 2 different pools of infected cells: epididymal epithelium for mir-202(T), and cells in epididymis lumen for mir-141(T). Unfortunately, the corresponding panel F from Fig4, which displays a single infected cell in the lumen, is difficult to conciliate with infection level results.

The authors should first demonstrate (picture in Fig4F) that there are not just very little numbers of infected cells in the lumen (maybe replace the pic 4F with the right panel in Figure S6 (mouse #2 epididymis)), and also perform a quantitative analysis of infected cells in immunohistochemistry labelling in order to explain the viral titer results (2x141(T) = 2x202(T) in epididymis at 12 dpi). Then the authors should explain why the infection level in epididymis 2xscr(T), which should display infected cells both in lumen and epithelium of epididymis, is not higher than 2x-202(T) (epithelium only) and 2x-141(T) (lumen only)?

The authors should characterize the cell type(s) present in the epididymis lumen: are they macrophages or germ cells? Macrophages could originate from epididymal interstitium and/or rete testis, when germ cells from testis only.

Third, if the cells present in epididymis lumen in mir-141(T) infected mice are from testis origin, they should be seen in 2xscr(T) too; Authors should show pictures of these cells and quantify.

Minor points:

1) Figure 1E and 1F: The authors should discuss and give some hypothesis on why the transition between the 2 phases (day 6 and 9 for testis, day 6 for epididymis) are specifically associated with very high SD values.

2) Semen analysis is an important missing point of the study. The authors should clearly mention that semen analysis would have been very useful to directly assess the potential impact of their system on sexual transmission. They should explain why they did not do it here and mention in the discussion how their model could be useful to trace the origin of ZIKV in semen. They should also mention that it is already known that, in addition to testis and epididymis, other organs from the male reproductive tract are source for the virus in semen. Indeed, ZIKV in semen after vasectomy was reported both in mouse model (Duggal et al., 2017) and infected patients (Froeschl et al., 2017; Arsuaga et al., 2016 and Huits et al., 2017). Moreover Zika virus was already reported in seminal vesicles, vas deferens and prostate of infected mice. The authors should highlight the value of their system in this context, or what could be done to improve it. They should mention if ZIKV antigen labeling was observed in spermatozoa as already reported in mouse model.

3) Once improved (see major points 4), the dual-route of epididymal infection will be a very interesting point and should be discussed further. The authors should mention what could be the impact on global MRT infection (could impact other MRT organs?), on virus persistence in male reproductive tract, and further. The authors should also quote and discuss a recent paper reporting similar cross-contamination hypothesis between testis and epididymis during SIV infection (Houzet et al, 2018).

4) The authors should mention in the discussion that we are in a AG129 mouse model, and that

such a sequential interstitial/tubules infection was not described in ZIKV infected human testis yet.

5) Authors should clearly indicate in Table S3 when deletion impact mir-202, mir-141 or both target sites.

6) page 3, line 52: ZIKV RNA in semen detected for up to one year post infection (Barzon et al., 2018).

7) page 13, line 180: before to mention escape mutations point, the authors should clearly explain to what corresponds the 2 peaks of infection observed in the testis (Fig 1E) at the immunohistochemical level. If the first peak is corresponding to interstitial infection and the second to seminiferous tubules, then the authors should show (see major point 2) and mention what is happening to interstitial infection at day 12.

8) page 13, line 185-186: please comment on why insertion of targets for mir-449a-5p did not affect virus replication.

9) Page 14, Title Fig 4: delete of in "...infection of and replication..."

10) Figure 4: red and magenta arrows are not easy to differentiate; chose other colors. The authors should include pictures showing distribution of ZIKV antigen distribution in epididymal interstitium of 2x141(T) mice at 3 dpi and 12 dpi. If available, graph with viral titers in the epididymis tissue of mice at 3dpi should be included.

11) Page 18, line 252: replace "Fig. 3F" with "Fig. 4F"

12) Figure S1: in panel B, replace 2xgfp with 2xscr; please use bigger arrows to highlight the miRNA target insertion sites in the ZIKV-NS3m sequence; In the B legend, mention that wt sequence is on the top and mutants below.

Response to the reviewers

We thank the reviewers for insightful and helpful comments and recommendations.

Answers to reviewers' comments.

Reviewers' comments:

Reviewer #1 (Remarks to the Author):

Major issues

1- In figure 1, the virus titer curve for epididymis is truncated and does not follow the same time-course as the other tissues. Various assumptions are made later in the manuscript that are based on limited observations of virus titer in the epididymis. It would be helpful if this tissue were evaluated at the same times as the other tissues. It is assumed that the tissue titer declines in the epididymis as it does in the testis, but this may not be the case and should be evaluated and included in the curve.

Response:

As per reviewer #1 request we completed evaluation of 2×scr virus replication in the epididymis at various time points (please see an updated version of **Fig. 1**). We observed that the titer of 2×scr reached a maximum at 9 dpi and declined thereafter. The peak of 2×scr virus replication in the testis occurred at 12 dpi. These new results do not contradict the hypothesis of dual route (i.e., hematogenous/lymphogenous and excurrent testicular) dissemination of ZIKV to the epididymis.

In addition, we now provide a comparison of the replication kinetics and survival curves between 2×scr and infectious cDNA clone-derived wild type Paraiba_01/2015 strain of ZIKV (ZIKV-NS3m) (reference 22, PMID: 27555311) as requested by Reviewer #2 (please see an updated version of **Fig. 1**). Discussion of these new findings is provided in the revised section 'Results', Subsection 'Model to study the effects of miRNA targeting on ZIKV tissue tropism'.

2- It would be useful to show virus replication curves generated in cell culture in order to compare wt and modified ZIKVs in order to better compare the replication of these viruses over time.

Response:

Upon reviewer's request, we compared replication kinetics of the ZIKV-NS3m virus and all miRNA targeted viruses in Vero cells. No statistically significant differences in the growth rates were detected. Results of this experiment are presented in the panel C of revised version of **Fig. 2**.

Changes in the text:

The sentence "All miRNA-targeted viruses replicated at nearly similar levels in Vero cells reaching titer of $\sim 7.0 \log_{10}$ (pfu/mL) by day 3 post plasmid DNA transfection (**Fig. 2A**)" was changed as follows: "All miRNA-targeted viruses grew with similar kinetics in Vero cells (**Fig. 2c**)..."

3- Could an explanation be given to why mir-449a-5p may not have worked to suppress viral growth in the testis? I do think it is useful to include negative data in the manuscript, but there should be more information included. The information is so brief on this construct that it is questionable why it is included at all.

Response:

A paragraph addressing the inability of target for mir-449a-5p to affect ZIKV replication in the testis is provided in the revised Discussion: 'Despite relatively high level of expression in the testis, targeting of ZIKV for mir-449a-5p did not affect accumulation of 2×449a(T) in this organ. Analysis of published data indicated that, compared to other testis-expressed miRNAs used in our study, mir-449a-5p is the least expressed in spermatogonia (but not in spermatocytes and spermatids) (Fig. S14)³². In addition, mir-449a-5p is not expressed in the Sertoli cells⁵², while both mir-202-5p and mir-465a-3p are highly expressed in this cell type⁵¹. Therefore, the fact that Sertoli cells and spermatogonia are now established as the main targets of ZIKV in the testis^{17,18,20}, but these cells do not express (or express at very low levels) mir-449a-5p, may explain the failure of mir-449a-5p-targeting to inhibit ZIKV replication in the testis.'

Minor issues:

1- Page 6, line 96. Please include a discussion of the figures and subfigures in order. I.e, the first subfigure mentioned is 1G.

Response:

The order of subfigures was corrected.

2- Page 7, line 108-109. This figure is not specific to the testis, so please modify this section name. I would also suggest moving the current figure 2F to the 2D position and 2D to 2E and 2E to 2F, so the figure is in order in relation to time.

Response:

The order of subfigures was modified according to reviewer's request. Corresponding changes were made in the text to ensure logical progression of the data representation.

3- Page 10, line 161. In addition to the testis, there is also significant reduction in the brain titers. Please include some indication why this might be the case.

Response:

There was no significant reduction in the brain titer of the Te-2×202(T)-mut or Te-2×202(T)-stb mice. We only observed a significant reduction of the CNS titers for clones which were targeted for brain-specific miRNA [2×9(T) and 2×124(T)]. These results were anticipated based on our previously published work (reference 43, 61), and we do not feel that additional explanations are necessary.

4-Page 18, line 270. Please include these abbreviations earlier in the manuscript where they first occur.

Response:

Corrected.

5- Page 20, line 301, suggested change to "...from Te-2x202/141(T)-stb-infected mice..."

Response:

We believe that the proposed change will be confusing, since it implies that some mice were specifically infected with a virus isolated from Te-2x202/141(T)-stb mice, which was not the case. Mice were infected only with 2x202/141(T) virus. Therefore, no changes were made.

6- Page 26, line 403 and throughout. I prefer not to see references to the figures in the discussion and suggest removing these.

Response:

We reasoned that providing references to the figures in the discussion will help the reader to quickly identify the results to support each particular statement in the discussion, without the need to read each result's section in full details. For this reason, we prefer to keep some references to the critical figures in the discussion (although total number of references to figures was reduced).

7- Page 35, lines 593-594, 605-606. Many different virus challenge doses were used. Please provide justification why these weren't kept consistent.

Response:

Justification for using different challenge doses was provided in the revised Methods section as follows:

'The 10⁵ pfu dose was selected to be consistent with the dose which is used in the monkey immunogenicity study (see below)'.

Reviewer #2 (Remarks to the Author):

This is a very well designed study by investigators with expertise in vaccine studies. The authors use the approach of targeting miRNA to achieve selective restriction of ZIKV replication in the testis and epididymis. They show that a combined co-targeting of ZIKV genome for the CNS-, testis- and epididymis-specific miRNAs restricts ZIKV infection of these organs, but does not impair the development of humoral immunity to ZIKV infection in mice. Strengths of the study include very strong data and innovative experimental approach to investigate ZIKV trafficking in the epididymis that can also be utilized for other models and other viruses. However, there are major concerns that the authors did not address.

- The use of AG129 mice is a significant limitation of the study and should be addressed. Being deficient in both type I and type II IFN receptors makes it a very artificial model and severely diminishes the predictive capacity of their results. IFNAR in particular allows uninfected cells to be alerted and mount an ISG response before coming in contact with the virus. ZIKV NS5 targets STAT2 for degradation in infected cells only,

thus intact IFNAR is still important for dissemination/trafficking. Its very unlikely that the trafficking of the virus would occur the same way within the same cells in an immunocompetent model, regardless of STAT2 antagonism by ZIKV NS5.

- Use of AG129 mice limits the predictive capacity of their approach in humans and nonhuman primates – in an immunocompetent scenario, many cells in the MRT may not be readily permissive to ZIKV infection, and therefore trafficking of virus through MRT would be different. Interpretation of ZIKV trafficking through the MRT may only be relevant to AG129 mice. These issues should be addressed by validating some of the experiments in immunocompetent mice treated with antibody to IFNs during infection to facilitate first wave of infection and tissue invasion of the virus. Addressing these limitations would strengthen the manuscript overall.

Response to both points:

We agree with Reviewer #2 that results obtained in immunodeficient mice cannot be directly applied to predict outcome of viral infection in the immunocompetent host. To acknowledge this limitation, we modified both the title and abstract to emphasize that presented results were obtained using immunodeficient model of Zika virus infection. We also added a discussion on the limitations of this model for interpretation of Zika virus dissemination in humans. Please see the paragraph starting with words ‘In the context of vaccine research...’.

Regarding the proposed experiments in immunocompetent mice treated with antibody to disrupt IFN signaling pathway, we believe that adaptation of such approach would add too many variables that will be difficult to control to achieve the goals of our current study. The variables inherently associated with such approach (that would potentially complicate our comparative virus analyses and make it difficult to interpret them) may include: (1) a route of antibody administration; (2) an optimal dose of antibodies to saturate the target (suggested to be a very large bolus injection, ref. PMID:17115899); (3) levels of antibody extravasation and penetration (if any) in a given tissue compartment (i.e, circulation/blood vs. testicular and/or epididymal interstitium vs. testicular tubular and/or epididymal ductal compartment; (4) timing of antibody administration and repeated dosing before and/or during ZIKVs infection (arbitrarily chosen in one report (see PMID: 28068342) at day -1, +1, and +4, which resulted in a high variability of observed outcomes); (5) target saturation/half-life and distribution of injected antibody in the MRT (especially in the immunoprivileged testis and epididymis (see PMID:2372399 and 24954222); (6) efficiency of binding and interferon/receptors neutralization by antibody in different cellular and structural compartments comprising the testis and epididymis. Thus, given a complexity of our experimental design with a large number of different viral constructs to be compared in a consistent manner and monitoring kinetics of their replication in the MRT, we could not afford to control for all variables mentioned above. Thus, we believe that the use of a well-established model to study Zika virus pathogenesis (see PMID:28148798) to achieve the goals of current study is sufficiently justified.

(2) - The NS5 of ZIKV inhibits STAT2 activation only in humans, but not mice (see PMID: 27212660). Therefore, immunocompetent mice in general (and mice treated with antibody to IFNs at the time when the injected antibody has been diluted out of circulation) are significantly more resistant to ZIKV infection as compared to humans. This means that ZIKV dissemination in the MRT of immunocompetent mice will not reflect the real ZIKV dissemination in the susceptible host such as human.

The only viable alternatives to study ZIKV dissemination in the MRT would be to use monkey model of infection or to use recently described hSTAT2 KI mice and mouse adapted strain of ZIKV (PMID:29746837). Both these models were briefly discussed in a revised version of the 'Discussion'. However, investigation of ZIKV dissemination in the MRT of any of these models is beyond the scope of presented manuscript.

- Discussion: Although their data showing protection of C/3'NCR-scr immunized AG129 mice to wt ZIKV challenge is compelling, however considering the limitation of the animal model, do the authors expect this type of protection to occur in other models? Authors should include their thoughts in the discussion.

Response:

We agree with Reviewer #2 that AG129 mice is not an adequate model for evaluation of vaccine immunogenic characteristics. To address this concern, we validated immunogenicity of the C/3'NCR-mir(T) virus in immunocompetent host (rhesus macaques). The findings of this new study are now presented as new paragraph added at the end of the section 'Results', subsection 'Application of simultaneous co-targeting of viral genome for CNS-, testis- and epididymis-specific miRNAs for the development of live-attenuated ZIKV vaccine'.

- At many places, the authors should clarify the statement by including either 'AG129' or 'immunocompromised mice'. For instance, line 368 should probably state "protective immunity in AG129 mice". Also the title should include '... virus dissemination in immunocompromised mice'

Response:

Line 368 was modified to 'protective immunity in AG129 mice and in immunocompetent primate host.'

The title was changed to – '**Controlled Zika virus dissemination in the testis and epididymis of immunodeficient mice: rational vaccine development**'. This new title specifically emphasizes that study was performed using immunocompromised mice.

- In some experiments comparison of 2xscr replication with wild-type virus would be helpful- specially replication in the testes and epididymis?

Response:

We compared replication kinetics of 2xscr to the parental ZIKV-NS3m virus in Vero cells and in all mouse organs tested in this study. These results are now reported in the updated version of **Fig. 1** and **Fig. 2**.

Reviewer #3 (Remarks to the Author):

Tsetsarkin et al. used miRNA target sites insertion in ZIKV genome to restrict its replication in selected organs and elucidate the routes of ZIKV infection in the male reproductive tract. Insertion of mir-202 (testis specific) and mir-141 (epididymis specific) target sites prevented infection of seminiferous tubules in testis and epithelial cells in epididymis respectively. They also observed that epididymis infection occurred via two routes, blood-associated and testicular. Finally, they generated a virus with restricted replication in CNS, testis and epididymis but a full humoral immunity potential. This study is interesting and brings convincing results for specific restriction of virus replication using miRNAs and promising data for a potential use for the development of a safe live-attenuated ZIKV vaccine. However, the part concerning the dual-route infection of epididymis is very weak and should be improved.

Major points:

1) Authors should add a minimum of miRNA quantification data and not depend fully on published data knowing that miRNA expression level can display important differences between individuals. It would be good to have at least mir-202 quantified in testis samples from the 12 mice corresponding to 2x202(T)-stb and 2x202(T)-mut. This could bring interesting details on: 1) mir-202 level variation between individuals. 2) Is there a negative correlation between miR-202 and virus levels in stb samples? 3) Comparing mir-202 levels in stb and mut samples could indicate whether escape mutants are associated with higher or lower level of selective pressure. At the very least, if frozen samples are no more available for miRNA quantification, the authors should discuss the parameters affecting the level of miRNA mediated selection pressure and/or emergence of escape mutants.

Response:

Unfortunately, the tissue samples from mice infected with miRNA targeted viruses that could be suitable for miRNA quantification are no longer available. However, we compared miRNA expression levels in the brain, epididymis and in testis isolated from 3 uninfected animals using deep sequencing analysis. We did not observe any substantial variation in the expression values for each miRNA of interest between individual mice. The results of this analysis are now presented in the updated **Fig. 2, Fig. 4, and Fig.S10**. We agree that an individual variability in the levels of miRNA expression can affect the miRNA mediated selective pressure and/or emergence of escape mutants. To address this, the following sentences were added to the discussion (please see the 1st paragraph): 'It appears that the emergence of the escape mutants is a stochastic process that likely depends upon individual and temporal variations in the level of expression of a specific miRNA and the efficiency of interactions of the miRNA silencing machinery with targets inserted into the virus. Thus, suboptimal levels of expression of specific miRNA in a given cell and a prolonged access of the virus to that cell can favor the emergence of escape mutants'

2) Figure 3 and Figure 4: characterization of interstitial labeling at day 12pi is missing in order to close the story of biphasic course of infection at the immunohistochemistry level. The authors should add pictures and quantitative analysis of interstitial labeling (immunohistochemistry) for testis and epididymis at day 12 pi.

Response:

Although each panel for 12 dpi in **Fig. 3** and **Fig. 4** already shows the absence of ZIKV labeling for non-structural protein (indicative of the absence of replicating virus), we now provide additional figures to highlight the clearance of replicating ZIKV from the testicular interstitial compartment (see **Supplemental Fig. S3**) as well as from the epididymal interstitial compartment (see **Supplemental Fig. S6**) at this time point. We specifically highlighted the interstitial compartments at high magnifications by a green overlay so a complete absence of ZIKV labeling can be clearly seen and appreciated. We believe that no quantification of labeling is necessary in this case.

The references to these new figures were provided in the text.

'By 12 dpi, both viruses were cleared from the interstitium (**Fig. S3**)' and

'Immunohistochemical examination showed that at 12 dpi, both 2x202(T) and 2xscr viruses were cleared from the interstitium (**Fig. S6**)'

3) Seminiferous tubules are composed of germ cells and somatic Sertoli cells, 2 very distinct cell lineages with little in common. Therefore, it would be important to precise whether mir202 is expressed in Sertoli or germ cells as it is unclear from the quoted references with discrepant results. This would be most useful for further approaches using mir202 (eg for viruses with different cell tropism) and may also shed light on the sequence of infection in the seminiferous tubules (eg Sertoli or germ cell mediated). For instance ISH on whole tissue or PCR on isolated cells could be performed.

Response:

We appreciate the reviewer's comment and agree that miRNA targeted viruses can be very useful and informative to precisely map the sequence order of events of ZIKV dissemination in the seminiferous tubules. To validate our results regarding the role of mir-202 in inhibition of 2x202(T) virus replication in seminiferous tubules, and to indirectly support hypothesis of dual route of dissemination of ZIKV into epididymis, we performed an in-situ hybridization experiment using probes for mir-202 and mir-141 in the mouse testis and epididymis. These new results are now presented in the **Supplemental Fig. S4** and clearly demonstrate that mir-202 (but not mir-141) is expressed in the most cell types residing within the seminiferous tubules, including the Sertoli cells and germ cells. This is in perfect agreement with previously published data from isolated cells (please see supplemental data in the references 51 (PMID:20467044) and 32 (PMID27998933)).

4) The weakest point of the paper is from far the demonstration of the dual-route (blood and testis) of epididymis infection.

Response:

1) To justify the weakness in our interpretation(s) of a particular experimental result, we would expect to see an alternative explanation(s) for the data presented in our manuscript. Alternatively, an internal inconsistency in our results [which could not be resolved on its own (see below)] could have been presented. Unfortunately, these were not clearly stated/indicated in the provided comments. Nevertheless, the skepticism of the Reviewer #3 about “the dual-route (blood and testis) of epididymis infection” led us to come up with and test our own alternative explanation for observed behavior of miRNA targeted viruses in the MRT. We believe that it is possible that attenuation of the 2×202/141(T) virus in the epididymis was not miRNA-mediated, but it was attributed to unknown phenomena associated with a particular sequence inserted into the 3’NCR of the 2×202/141(T). This would invalidate the ‘dual route dissemination’ hypothesis proposed to explain ZIKV invasion into epididymis. However, if the miRNA-mediated mechanism of attenuation of the 2×202/141(T) is valid, then any ZIKV clone simultaneously targeted for other testis- and epididymis-specific miRNAs should also be attenuated in the testis and epididymis, as long as selected miRNAs independently restrict ZIKV replication in the cells of testicular seminiferous tubules and in the epididymal epithelium. Stated in another way, it is possible that the attenuation of the 2×202/141(T) in the epididymis was not a unique event, but was a common consequence of simultaneous restriction of both hematogenous/lymphogenous and excurrent testicular routes of infection by dual miRNA targeting.

Due to the limitations on allowed length of the manuscript, the detailed description of experimental results which validated the dual route dissemination hypothesis using targets for additional testis- and epididymis-specific miRNAs are provided in the updated version of Supplemental materials (See **Fig S10**, and **Tables S4** and **S5**). Only a brief description of the rationale for new experiments and general conclusion was provided in the main body of the paper (See last paragraph in the section ‘**ZIKV can invade epididymis via excurrent testicular route of infection**’).

2 - General considerations:

The main concern of the reviewer #3 regarding dual dissemination route hypothesis is an apparent ‘discrepancy’ between infectious titers of different viruses in the epididymis and corresponding immunohistochemical staining of these viruses in the organ. However, these discrepancies are resolved considering that majority of the virus in the lumen of epididymis can exist in a cell-free form (which was shown to be the case for ZIKV in the mouse ejaculate; see reference 21 [PMID: 30070988]). The cell-free virus cannot be visualized by immunohistochemical analysis, which resolves the discrepancy. The reviewer is also interested in performing detailed analysis of different cell types which stained positive for ZIKV antigen in the lumen of epididymis for different viruses. This is an interesting scientific question, but it will not provide critical missing information that can help prove or disprove the dual dissemination route hypothesis. Moreover, this question has very limited relevance to the ZIKV vaccine applications described in the manuscript. However, in order to address the question regarding the potential role of the cells that stain positive for ZIKV in the lumen of epididymis in the general form, we decided to evaluate ZIKV antigen distribution in the mice infected with

2x202/141 virus. We proposed that 2x202/141 virus is supposed to be restricted in ability to infect epididymis by both excurrent testicular and hematogenous routes of infection. Therefore, there should be no ZIKV positive cells in the lumen of epididymis infected with this virus (regardless of which particular type of cells becomes infected by different miRNA-targeted ZIKV in the epididymis). In our analysis, we did not observe ZIKV-positive staining in any cells in the lumen of all analyzed organs (n=4) (see updated **Fig. 3** and **4**). This indicates that dual targeting of ZIKV for mir-202 and mir-141 prevents infection of any cells in the lumen (regardless of what particular type of cells that might be). This novel observation additionally supports the dual dissemination route hypothesis. It is also particularly important in the context of the ZIKV vaccine research applications, allowing development of a live ZIKV vaccine that does not infect testis and epididymis.

3- Point by point **Responses:**

Figure 4A shows that 2x141(T) virus level in epididymis at 12 dpi is the same as 2x202(T); however, immunohistochemical analysis clearly demonstrates that this corresponds in fact to 2 different pools of infected cells: epididymal epithelium for mir-202(T), and cells in epididymis lumen for mir-141(T). Unfortunately, the corresponding panel F from Fig4, which displays a single infected cell in the lumen, is difficult to conciliate with infection level results. The authors should first demonstrate (picture in Fig4F) that there are not just very little numbers of infected cells in the lumen (maybe replace the pic 4F with the right panel in Figure S6 (mouse #2 epididymis)),

Response:

Upon reviewer's request the panels were replaced.

and also perform a quantitative analysis of infected cells in immunohistochemistry labelling in order to explain the viral titer results (2x141(T) = 2x202(T) in epididymis at 12 dpi).

Response:

There is no apparent contradiction between titers of 2x141(T) and 2x202(T) viruses at 12 dpi and the corresponding immunohistochemical analysis of these viruses in the epididymis. It is most likely that the majority of 2x141(T) virus in the lumen of epididymis exists in a cell-free form which could have been transported from the testis. Occasional ZIKV positive cells could be sloughed spermatogenic precursors cells or leukocytes. The experiments proposed by reviewer #3 characterizing cell populations in the lumen of epididymis would be redundant in the light of the recently published study [see reference 21(PMID:30070988)] which addressed exactly this question. This study (ref 21) showed that in the ejaculate ZIKV exists mostly in cell-free form and ZIKV positive cells in lumen of the epididymis are sloughed spermatogenic precursors cells or, less likely, leukocytes.

Amendment to the text: the following sentence was added to the text providing clarification for the high titer of 2x141(T) in the epididymis and adding a new reference

to the above-mentioned study addressing the issue of luminal contest of ZIKV infected epididymis .

‘While this manuscript was under consideration, a new study identified these ZIKV positive epididymal luminal cells as spermatids (Prm2+) which have been sloughed from the testicular seminiferous tubules and transported to the epididymis via excurrent ducts or/and they could be infected luminal leukocytes²¹,

Then the authors should explain why the infection level in epididymis 2xscr(T), which should display infected cells both in lumen and epithelium of epididymis, is not higher than 2x-202(T) (epithelium only) and 2x-141(T) (lumen only)?

Response:

Considering high variation (10 folds) observed between titers of the same virus in the epididymis samples isolated from different mice, it is quite expected that relatively small differences (2-folds) due to restriction of a particular route of dissemination were not detected in our study using relatively small sample size (n=4 or 5).

Following sentences were added to the text: “Restriction of individual dissemination routes for viruses 2×202(T) (testicular) or 2×141(T) (hematogenous) suggests, that compared to 2×scr, there should have been some reduction in the titer of each of these viruses in epididymis. However, this was not detected in our experiments. We speculate that variation between viral titer in the epididymis of different mice at 12 dpi was substantially greater compared to relatively small differences (2-folds), which are expected due to a restriction of each of the two dissemination routs. Also, considering that ZIKV reaches maximum titer in the epididymis slightly faster than in testis, we think that restriction of hematogenous pathway by mir-141-3p targeting would have more pronounced effect on attenuation of the 2×141(T) virus in the epididymis, if it was assessed during earlier times post infection (6 or 9 dpi) compared to that observed at 12 dpi.”

The authors should characterize the cell type(s) present in the epididymis lumen: are they macrophages or germ cells? Macrophages could originate from epididymal interstitium and/or rete testis, when germ cells from testis only.

Response:

See discussion above: **part 2 - General considerations.**

Third, if the cells present in epididymis lumen in mir-141(T) infected mice are from testis origin, they should be seen in 2xscr(T) too; Authors should show pictures of these cells and quantify.

Response:

The infected cells were often observed in the lumen of epididymis infected with 2xscr(T). In the original version of the manuscript we did not consider intraluminal staining as a significant evidence for or against dual dissemination mechanism, and randomly selected image that didn’t have ZIKV-positive cells in the lumen. In the revised version of paper this image was replaced with image of epididymis lumen infected with 2xscr(T) which contains numerous ZIKV-positive cells (see revised **Fig. 3 and 4**).

Minor points:

1) Figure 1E and 1F: The authors should discuss and give some hypothesis on why the transition between the 2 phases (day 6 and 9 for testis, day 6 for epididymis) are specifically associated with very high SD values.

Response:

We believe that a structural tissue barrier and/or switching the host cells between 2 phases is responsible for high variability of 2×scr titer.

The sentence “This may explain a high variability of 2×scr titers during transition between the 2 phases of replication (6-9 dpi for testis, 6 dpi for epididymis).” was added after the sentence “A biphasic profile of 2×scr replication in both organs suggests that the virus encountered a structural tissue barrier which slowed down its dissemination, and/or switched the host cells to replicate more efficiently”.

2) Semen analysis is an important missing point of the study. The authors should clearly mention that semen analysis would have been very useful to directly assess the potential impact of their system on sexual transmission. They should explain why they did not do it here and mention in the discussion how their model could be useful to trace the origin of ZIKV in semen.

Response:

We agree that analysis of ZIKV dissemination in the MRT should be culminated and closed by a study of ZIKV in the semen and sexual transmission experiments. However, these experiments could not be properly conducted without developing the model for selective inhibition of ZIKV replication in the testis and/or epididymis first: the two organs of the MRT located upstream of the sperm flow and most susceptible to ZIKV infection. However, considering the amount of additional experiments which would be required to address these questions, it is quite apparent that such work cannot be performed in a short period of time and published as a single manuscript.

The explanation why we focused only on the testis and epididymis is given in the last paragraph of revised introduction (please see the sentences ‘In immunodeficient mice, ZIKV primarily targets the testis and epididymis, and to a lesser extent, the prostate and seminal vesicles¹⁷⁻²¹. In this study, we used a host microRNA (miRNA) targeting approach to trace routes of ZIKV dissemination in the MRT of mice. Since ZIKV generated in one part of the MRT can be transported to the other parts located downstream of excurrent flow, we focused only on the ZIKV interaction with major upstream organs, namely, the testis and epididymis.’)

They should also mention that it is already known that, in addition to testis and epididymis, other organs from the male reproductive tract are source for the virus in semen. Indeed, ZIKV in semen after vasectomy was reported both in mouse model (Duggal et al., 2017) and infected patients (Froeschl et al., 2017; Arsuaaga et al., 2016 and Huits et al., 2017). Moreover Zika virus was already reported in seminal vesicles, vas deferens and prostate of infected mice. The authors should highlight the value of their system in this context, or what could be done to improve it. They should mention if ZIKV antigen labeling was observed in spermatozoa as already reported in mouse model.

Response:

The following paragraph was added to a revised Discussion:

'ZIKV has also been detected in the semen of vasectomized men^{55,56} and mice¹⁹ implicating vas deferens, seminal vesicles, and the prostate as important organs potentially contributing to ZIKV shedding into semen. Future work should be focused on understanding of a relative contribution of the individual MRT organs to the ZIKV infectivity, persistence, and sexual transmission. The miRNA-targeting approach for restriction of ZIKV replication in each component of the MRT individually, and viruses with altered tissue tropisms described here, represent valuable experimental tools to address these and other outstanding questions regarding ZIKV infection of the MRT⁴⁹,

3) Once improved (see major points 4), the dual-route of epididymal infection will be a very interesting point and should be discussed further. The authors should mention what could be the impact on global MRT infection (could impact other MRT organs?), on virus persistence in male reproductive tract, and further. The authors should also quote and discuss a recent paper reporting similar cross-contamination hypothesis between testis and epididymis during SIV infection (Houzet et al, 2018).

Response:

The impact of proposed dual-route of epididymal infection on the global ZIKV dissemination and persistence in the MRT is relatively self-explanatory. We decided to omit such discussions due to the length constraints of the manuscript. However, we discussed findings of (Houzet et al, 2018) in the context relevant to AG129 mice model for evaluation of viral dissemination in the MRT.

The following paragraph was added to a revised Discussion:

'Interestingly, viral dissemination between testis and epididymis of NHPs was demonstrated for another sexually transmitted pathogen - simian immunodeficiency virus⁵⁴, suggesting that this route of spreading could be utilized by ZIKV in primates as well.'

4) The authors should mention in the discussion that we are in a AG129 mouse model, and that such a sequential interstitial/tubules infection was not described in ZIKV infected human testis yet.

Response:

We discussed limitations of AG129 mouse model for the research of ZIKV pathogenesis in an additional paragraph of the revised Discussion:

'In the context of vaccine research, evaluation of the 'worst case scenario' outcomes of ZIKV infection using immunodeficient AG129 model can ensure vaccine safety for all recipients, whose immunological status could be unknown. However, the reliance on this model to study pathogenesis of ZIKV could generate results that might not be fully relevant for an immunocompetent host whose cells might be less permissive to infection. Obviously, ZIKV infection and dissemination in human testis and epididymis remains to be fully elucidated. It would be, therefore, important to evaluate ZIKV dissemination routes in the MRT of non-human primates (NHPs) or in the MRT of recently developed immunocompetent hSTAT2 KI mice using mouse adapted strain of ZIKV⁵³.'

In addition, we discussed novel findings of Matusali et al, 2018 showing ZIKV infection of human testis *ex vivo* in the revised Introduction.

'Human testicular tissue explants support active ZIKV replication, which occurs primarily in macrophages and germ cells¹⁶.

5) Authors should clearly indicate in Table S3 when deletion impact mir-202, mir-141 or both target sites.

Response:

The position of the detected nucleotides in the sequence of 2×202/141(T) virus and how it affects targets for mir-202 and mir-141 was provided in modified Table S3. Similar notations for new viruses 2×465a(T) and 2×202/200c(T) were provided in the new Table S4 and S5.

6) page 3, line 52: ZIKV RNA in semen detected for up to one year post infection (Barzon et al., 2018).

Response:

The phrase 'up to 9 months' was corrected to 1 year and reference to (Barzon et al., 2018) was added to the manuscript.

7) page 13, line 180: before to mention escape mutations point, the authors should clearly explain to what corresponds the 2 peaks of infection observed in the testis (Fig 1E) at the immunohistochemical level. If the first peak is corresponding to interstitial infection and the second to seminiferous tubules, then the authors should show (see major point 2) and mention what is happening to interstitial infection at day 12.

Response:

See response to major point#2

8) page 13, line 185-186: please comment on why insertion of targets for mir-449a-5p did not affect virus replication.

Response:

A paragraph addressing the inability of target for mir-449a-5p to affect ZIKV replication in the testis is provided in the revised Discussion: 'Despite relatively high level of expression in the testis, targeting of ZIKV for mir-449a-5p did not affect accumulation of 2×449a(T) in this organ. Analysis of published data indicated that, compared to other testis-expressed miRNAs used in our study, mir-449a-5p is the least expressed in spermatogonia (but not in spermatocytes and spermatids) (Fig. S14)³². In addition, mir-449a-5p is not expressed in the Sertoli cells⁵², while both mir-202-5p and mir-465a-3p are highly expressed in this cell type⁵¹. Therefore, the fact that Sertoli cells and spermatogonia are now established as the main targets of ZIKV in the testis^{17,18,20}, but these cells do not express (or express at very low levels) mir-449a-5p, may explain the failure of mir-449a-5p-targeting to inhibit ZIKV replication in the testis.'

9) Page 14, Title Fig 4: delete of in "...infection of and replication..."

Response:
Corrected

10) Figure 4: red and magenta arrows are not easy to differentiate; chose other colors.

Response:
Corrected.

The authors should include pictures showing distribution of ZIKV antigen distribution in epididymal interstitium of 2x141(T) mice at 3 dpi and 12 dpi. If available, graph with viral titers in the epididymis tissue of mice at 3dpi should be included.

Response:
New figure showing ZIKV antigen distribution of the 2x141(T) virus at 12 dpi in epididymis was provided in revised supplementary material. New graph showing viral titers in the epididymis of mice at 3dpi was also provided as a new panel in the revised **Fig. 4**. This new graph shows no difference between titer of 2x141(T) and all other viruses. That makes obtaining new pictures showing distribution of 2x141(T) antigen at 3 dpi redundant.

11) Page 18, line 252: replace “Fig. 3F” with “Fig. 4F”

Response:
Corrected

12) Figure S1: in panel B, replace 2xgfp with 2xscr; please use bigger arrows to highlight the miRNA target insertion sites in the ZIKV-NS3m sequence; In the B legend, mention that wt sequence is on the top and mutants below.

Response:
All three points were corrected

Reviewers' Comments:

Reviewer #1:

Remarks to the Author:

It appears the authors have suitably addressed all of the reviewer's concerns. I would recommend publication of this improved manuscript. The material covered and the methods used provide some interesting and novel results and would benefit this area of research.

Reviewer #2:

Remarks to the Author:

In this re-submission, the authors have addressed most of the concerns raised by all three reviewers. This reviewer is satisfied by new changes made to the manuscripts including new title and discussion to reflect the strengths and the weaknesses of the model used. Addition of new experiment using macaque model also has strengthened the study.

Saguna Verma

Reviewer #3:

Remarks to the Author:

Most of the points have been correctly answered in this revised manuscript and the addition of new results has improved its quality.

Minor points:

- 1) Figure 2a, graph title: correct "espression"
- 2) Graph title in Figure 2a should be the same format as 4a; graph titles in 4b and 4c should be the same format as 2d-h
- 3) page 14, line 283-286: not clear now to explain this strange result in the brain. Maybe better to put back the text from the original version of the article (page 23, line 336 to 340).

Response to the reviewers

We thank the reviewers for insightful and helpful comments and recommendations.

Answers to reviewers' comments.

Reviewer #3 (Remarks to the Author):

Most of the points have been correctly answered in this revised manuscript and the addition of new results has improved its quality.

Minor points:

1) Figure 2a, graph title: correct "espression"

Response:

Corrected.

2) Graph title in Figure 2a should be the same format as 4a; graph titles in 4b and 4c should be the same format as 2d-h

Corrected.

3) page 14, line 283-286: not clear now to explain this strange result in the brain. Maybe better to put back the text from the original version of the article (page 23, line 336 to 340).

Response:

According to reviewer's request the phrase on lines 283-286 'suggesting a redundancy of the CNS-specific miRNAs (mir-124 and mir-9) for restriction of C/3'NCR-mir(T) virus neurotropism.' was replaced with that from the original version of the paper 'indicating that non-miRNA-mediated attenuation of ZIKV due to dCGR formation in a combination with insertion of heterologous sequences into the 3'NCR is sufficient to restrict ZIKV neuroinvasiveness.'